



# Patterns of carbon processing at the seafloor: the role of faunal
# and microbial communities in moderating carbon flows
C. Woulds[1], S. Bouillon[2], G. L. Cowie[3], E. Drake[1], Jack J. Middelburg[4, 5], U. Witte[6]
[1]water@leeds, School of Geography, University of Leeds, Leeds, LS2 9JT, UK
[2]Department of Earth and Environmental Sciences, KU Leuven, Leuven, Belgium
[3]School of GeoSciences, University of Edinburgh, West Mains Road, Edinburgh, EH9 3JW, UK
[4]Royal Netherlands Institute of Sea Research (NIOZ-Yerseke), PO Box 140, 4400 AC Yerseke, The
Netherlands
[5]Department of Earth Sciences, Utrecht University, PO Box 80021, 3508 TA Utrecht, The Netherlands
[6]Institute of Biological and Environmental Sciences, Oceanlab, University of Aberdeen, Aberdeen, AB41 6AA,
UK
*Correspondence to:* C. Woulds (c.woulds@leeds.ac.uk)



**Abstract**
Marine sediments, particularly those located in estuarine and coastal zones, are key locations for the burial of
organic carbon (C). However, organic C delivered to the sediment is subjected to a range of biological C-cycling
processes, the rates and relative importance of which vary markedly between sites, and which are thus difficult
to predict.
In this study, stable isotope tracer experiments were used to quantify the processing of C by microbial and
faunal communities in two contrasting Scottish estuarine sites: a subtidal, organic C rich site in Loch Etive with
cohesive fine-grained sediment, and an intertidal, organic C poor site on an Ythan estuary sand flat with coarse-
grained permeable sediments.
In both experiments, sediment cores were recovered and amended with $^{13}$C labelled phytodetritus to quantify
whole community respiration of the added C and to trace the isotope label into faunal and bacterial biomass.
Similar respiration rates were found in Loch Etive and on the Ythan sand flat ($0.64\pm0.04$ and $0.63\pm0.12$ mg C
$m^{-2}h^{-1}$, respectively), which we attribute to the experiments being conducted at the same temperature. Faunal
uptake of added C over the whole experiment was markedly greater in Loch Etive ($204\pm72$ mg C $m^{-2}$) than on
the Ythan sand flat ($0.96\pm0.3$ mg C $m^{-2}$), and this difference was driven by a difference in both faunal biomass
and activity. Conversely, bacterial C uptake over the whole experiment in Loch Etive was much lower than that
on the Ythan sand flat ($1.80\pm1.66$ and $127\pm89$ mg C $m^{-2}$ respectively). This was not driven by differences in
biomass, indicating that the bacterial community in the permeable Ythan sediments was particularly active,
being responsible for $48\pm18\%$ of total biologically processed C. This type of biological C processing appears to
be favoured in permeable sediments. The total amount of biologically processed C was greatest in Loch Etive,
largely due to greater faunal C uptake, which was in turn a result of higher faunal biomass. When comparing
results from this study with a wide range of previously published isotope tracing experiments, we found a strong
correlation between total benthic biomass (fauna plus bacteria) and total biological C processing rates.
Therefore, we suggest that the total C cycling capacity of benthic environments is primarily determined by total
biomass.





## 1 Introduction

The burial of organic carbon in marine sediments is a key flux in the global carbon (C) cycle, linking the surface reactive C reservoirs to long term storage in the geological loop. In addition, organic detritus is the main food source for most benthic ecosystems, and its supply and cycling are thus important controlling factors for benthic ecology. Furthermore, the degradation of organic carbon (OC) in sediments usually drives their redox state, and together these determine nutrient regeneration rates and resupply to the water column. Estuarine sediments are particularly important locations for these functions. Of all marine benthic environments, estuarine (particularly fjordic) and shelf sediments host the largest proportion of marine sediment C burial (Berner, 1982; Duarte et al., 2005, Smith et al., 2015). The shallow water depths in estuaries result in the potential of benthic C burial and nutrient regeneration to control water column biogeochemistry and productivity (e.g. Middelburg and Levin, 2009). Therefore, there is a need to understand OC cycling and burial in marine sediments, and in estuarine sediments in particular.

Previous work has established that factors such as OC loading and degradation state, sediment grain size, and the time for which OC is exposed to oxygen before being buried below the oxycline combine to control the relative importance of remineralization and burial as a fate of C in marine sediments (Canfield et al., 1994; Mayer, 1994; Hedges and Keil, 1995; Hartnett et al., 1998). However, the pathways along which OC may travel towards burial or remineralisation must be elucidated in order to further our understanding of benthic C cycling and burial.

There are many processes to which OM arriving at the sediment surface, either of terrestrial origin delivered through riverine inputs or from surface phytoplankton production, may be subjected. First, a major fraction of fresh OC inputs may be fed upon by benthic fauna (Herman et al., 1999; Kristensen, 2001). Thus, C may be assimilated into faunal biomass, and may be transferred through benthic and/or pelagic food webs. Alternately, ingested sedimentary OC may survive gut transit and be egested back into the sediment, in which case it is likely to have been biochemically altered and physically re-packaged (e.g. Bradshaw et al., 1990 a, b; 1991 a, b; Woulds et al., 2012; 2014). In addition, at any trophic level of the food web, C may be metabolised and returned to the water column as $CO_2$. Further, during bioturbation many fauna transport OC through the sediment column, which may subject it to fluctuating redox conditions and accelerate decay, or sequester it at depth below the digenetically active zone (Aller, 1994; Sun et al., 2002). Secondly, deposited OC will be subject to microbial decay, and may thus be incorporated into microbial biomass, which itself may then progress through the foodweb, or may be returned to the water column as $CO_2$ through microbial respiration. In addition, it may be released as dissolved organic C (DOC) and re-incorporated into microbial and, subsequently, faunal biomass through the microbial loop (Pozzato et al., 2013 and references therein).

As the processes described above are all biologically driven, we will refer to them collectively as biological C processing. The relative importance of the different processes, in turn, will be referred to as the biological C processing pattern.

Isotope tracer experiments, in which organic matter labelled with an enriched level of a naturally uncommon stable isotope (typically $^{13}C$ and/or $^{15}N$) are an ideal tool to derive direct quantitative data on biological C





processing patterns and rates (Middelburg, 2014). Such experiments have been conducted in a wide range of
benthic environments, from estuarine sites (Moodley et al., 2000) to the deep abyssal plain (Witte et al., 2003 b),
from OC rich sediments (Woulds et al., 2007) to oligotrophic sites (Buhring et al., 2006 b), and from polar
regions (Gontikaki et al., 2011) to the tropics (Aspetsberger et al., 2007; Sweetman et al., 2010).
Many isotope tracer studies have found remineralisation by the entire benthic community (i.e. bacterial, meio-,
and macrofauna combined) to form the dominant fate of the OC supplied (e.g., Woulds et al., 2009; Gontikaki et
al., 2011c). It is reasonably well established that such benthic respiration rates are strongly controlled by
temperature (Moodley et al., 2005), and also respond to OC input (Witte et al., 2003 b) and benthic community
biomass (e.g. Sweetman et al., 2010)
However, considerable variations in carbon processing patterns and rates have been found between sites, with
considerable differences in, for example, the biomass pools into which OC is dominantly routed. Thus, some
studies have shown that OC uptake by foraminifera and/or bacteria can dominate in both the short and long term
(Moodley et al., 2002; Nomaki et al., 2005; Aspetsberger et al., 2007), and others have shown a more prominent
role for macrofauna (Witte et al., 2003 a). In some cases macrofaunal uptake can even be equal to total
respiration (Woulds et al., 2009). Trends in faunal OC uptake are usually strongly determined by trends in the
biomass of different faunal groups (e.g. Woulds et al., 2007; Hunter et al., 2012b), although this is not always
the case. For example, in sandy subtidal sediments, Evrard et al. (2010) found that more microphytobenthos C
was consumed by meiofauna than by macrofauna, despite the lower biomass of the former. In cohesive
sediments from a deep fjord, however, the opposite pattern was observed, when macrofaunal foraminifera
ingested less OC than expected based on their importance in terms of biomass (Sweetman et al., 2009). This was
thought to be due to their relatively deep dwelling lifestyle, suggesting they were not adapted for rapid feeding
on freshly deposited OM. Thus, the ecology and community structure of any site is thought to exert significant
control on its biological C processing pathways and rates. Furthermore, the examples given above illustrate how
the extreme variability in the abundance and characteristics of organisms found at seafloor sites throughout the
marine environment has resulted in the lack of a general understanding of how benthic communities impact
seafloor C cycling patterns and rates.
In a review of isotope tracer experiments carried out in marine sediments, Woulds et al. (2009) proposed a
categorisation of biological C processing patterns into three main types. 'Respiration dominated' sites were
defined as systems in which >75% of biologically processed C was found as respired $CO_2$, and this tended to
occur mostly in deep, cold, OM-poor sites with relatively low faunal biomass. 'Active faunal uptake' systems
were described as sites in which respiration was still the major fate of biologically processed C, but where
faunal uptake accounted for 10-25%. This pattern was found in shallower, more nearshore and estuarine sites,
which were richer in OM, and which hosted correspondingly higher benthic faunal biomass. A third category
labelled 'metazoan macrofaunal dominated' displayed an unusual pattern in which uptake by metazoan
macrofauna accounted for >50% of biological C processing, and was chiefly exhibited in a lower oxygen
minimum zone site on the Pakistan margin, where high OC concentrations and just sufficient oxygen supported
an unusually high macrofaunal biomass (an 'edge effect', Mullins, 1985). This categorisation allowed
predictions to be made regarding C processing patterns at a range of sites, but this ability was limited to the
types of benthic environment in which isotope-tracing experiments had been conducted to that date.



The previously proposed categorisation was limited in the types of benthic environments covered, and was
biased towards subtidal and deep-sea settings characterized by cohesive sediments. Therefore, a particular
environment missing in previous syntheses was coarse-grained, permeable sediments, such as are typically
found in coastal and shelf environments. One study in subtidal sandy sediments of the German Bight found
unexpectedly rapid C processing rates, and suggested a C processing pattern that was dominated by bacterial
uptake (Buhring et al., 2006 a). However, variation in results between different experiment durations implies
that it could not be used to propose an additional category. The result was however consistent with recent
findings that coarse-grained, permeable sediments are capable of more dynamic biogeochemical cycling than
was previously assumed from their generally low OC contents (Huettel et al., 2014). The rapid biogeochemical
cycling is driven by water flow over roughness on the sediment surface creating local pressure gradients, which
lead to advective exchange of porewaters. This introduces fresh organic substrates and electron acceptors into
the sediment, and removes metabolites, enhancing OC turnover (Huettel et al., 2014, and references therein).
Therefore, further investigation of biological C processing in previously understudied permeable sediments is
warranted.
Our study aimed broadly to investigate biological C processing rates and patterns in estuarine sediments. In
particular, we aimed to compare biological C processing in cohesive, fine-grained sediments with that in
permeable, coarse-grained sediments and to contrast the roles played by two communities with different
compositions and structures. We hypothesised that, in keeping with previous subtidal/shelf/fjordic sites, the
cohesive sediments would exhibit a C processing pattern dominated by respiration but with a marked role for
faunal uptake, while permeable sediments would exhibit rapid OC turnover, and an OC processing pattern
dominated by bacterial uptake. Further, we hypothesised that while faunal C uptake at the two sites would
necessarily involve different taxa, the overall contribution of fauna to biological C processing would be related
to their total biomass.
**2 Methods**
**2.1 Study sites**
Two sites were selected for study: one fine-grained, organic carbon-rich site in Loch Etive and a sandy site with
low organic carbon content in the Ythan estuary.
Loch Etive lies on the west coast of Scotland (Fig. 1). It is a glacier carved feature, 30 km long, and is divided
into three basins by two shallow sills at Bonawe and Connel (Fig. 1). The loch exhibits positive estuarine
circulation, with a strong outflow of freshwater in the surface 10m, and tidal exchange of seawater beneath (tidal
range is 2 m, Wood et al. 1973). Phytoplankton standing stock has been found to be relatively high (Wood et al
1973). This, combined with input of substantial amounts of terrestrial OC and the tendency of fine sediment to
be resuspended from the shallower areas and redeposited in the deeper areas (Ansell 1974) leads to relatively
OC rich sediments in the deep basins. The site chosen for this study lies at the deepest point (Airds Bay, 70 m)
of the middle basin of Loch Etive (Fig. 1). While the bottom water here is regularly renewed and is therefore
well oxygenated, the sediment has a relatively high oxygen demand, and sulphate reduction occurs within 5 cm
of the sediment-water interface (Overnell et al., 1996). The experiment was conducted during July 2004, at




which point the bottom water dissolved oxygen saturation was close to 100%. The sediment had a median grain
size of 21 m with 78 % fines (<63 m) and contained ~4.9wt % organic C (Loh et al., 2008). The benthic
community was dominated by ophuroids, with polychaetes and molluscs also being abundant (Gage 1972, C.
Whitcraft unpubl. data).
The Ythan estuary is a well-mixed estuary on the East coast of Scotland (Fig. 1), 20 km north of Aberdeen. It is
~8 km long, with a mean width of 300 m. The Ythan sand flat study site was located around halfway along the
estuary on an intertidal sand bar, and exhibited sandy, permeable and OC poor (~0.1 wt % organic C) sediments
(Zetsche et al., 2011b) which were subject to semi-diurnal tides and seasonal storms. The median grain size was
336 μm with 11% fines (<63 μm, varying through the year), and the sand is described as well sorted (Zetsche et
al., 2011 a). The study site was exposed at low tide, and covered by 1-2 m of water at high tide. The benthic
community was dominated by oligochaetes, with polychaetes, molluscs, nematodes and crustaceans also present
(Zetsche et al., 2012). The Ythan sand flat experiment was conducted during May 2008.

## 2.2 Isotope tracing experiments

The experimental setup varied slightly between sites, to account for the differences in their depth and sediment
grain size.

### 2.2.1 Loch Etive

Four replicate sediment cores (up to 50 cm depth, 10 cm i.d.) were collected and placed in a controlled
temperature laboratory set to the ambient temperature of 11°C. Phytodetritus labelled with $^{13}$C (~25%) was
added to the sediment surface of intact cores to give a dose of $1050 \pm 25$ mg C m$^{-2}$ (the standard deviation stated
is due to variation between replicate cores). The cores were then sealed and incubated in the dark for 7 days
(156 h). During the incubation, the oxygen concentration in core-top water was maintained by pumping the
water through an 'oxystat' gill, composed of gas permeable tubing submerged in a reservoir of 100%
oxygenated seawater (see Woulds et al., 2007), and monitored with Clark type electrodes. As the tubing used in
the oxystat gill was permeable to all gases there was the potential for loss of some $^{13}$CO$_2$ generated during the
experiment. However, the dissolved inorganic carbon (DIC) concentration difference between the incubation
water and oxygenated reservoir will have remained small, thus this effect is thought to be minor. Samples of the
overlying water were taken at 0, 24, 48, 72, 96, 120 and 144 hours after the introduction of the labelled
phytodetritus. These were preserved in glass vials without a headspace and poisoned with HgCl$_2$ for DIC and
$\delta^{13}$C- DIC analysis.
At the end of the incubation period, cores were sectioned at intervals of 0.5 cm up to 2 cm depth, then in 1 cm
sections up to 10 cm depth, and finally in 2 cm sections up to 20 cm depth. Half of each sediment slice was
sieved, with >300 μm (macrofauna) and 150-300 μm (meiofauna) fractions retained. The other half of each slice
was stored frozen in plastic bags. Sieve residues were examined under the microscope and all fauna were
extracted. Organisms were sorted to the lowest taxonomic level possible and preserved frozen in pre-weighed
tin boats and pre-combusted glass vials. Fauna from two of the four cores were allowed to void their guts before
preservation. This was achieved by allowing them to remain in dishes of Milli-Q water for several hours before
freezing.



### 2.2.2 The Ythan sand flat


Four replicate sediment cores were collected by pushing 25 cm diameter acrylic core tubes into the sediment at
low tide, and digging them out to obtain intact sediment cores 14-15 cm in length. These were returned to a
controlled temperature laboratory set to 11°C at Oceanlab, University of Aberdeen. Filtered Ythan estuary water
was added to each core to create a water column. A lid was placed on each core, leaving a headspace, with
exhaust ports open. Fully oxygenated conditions were maintained by gentle bubbling with air, except during
respiration measurements (see below). Lids were mounted with stirring disks, the rotation rates of which were
calibrated to generate appropriate pressure gradients to prompt porewater advection (Erenhauss and Huettel,
2004). The overlying water was changed daily. Isotopically labelled (34 % $^{13}$C) phytodetritus was added to the
water column and allowed to sink onto the sediment-water interface to give a dose of 753±9.4mg C m$^{-2}$. Twice
during the subsequent 7 days (immediately after phytodetritus addition and 5 days later) the respiration rate in
each core was measured. This involved filling the headspace in each core to exclude all air bubbles and sealing
all lids. Time series water samples were taken over the subsequent 24 h and preserved for $\delta^{13}$C DIC analysis as
described above. At the end of each respiration measurement, lids were removed and dissolved oxygen was
measured by Winkler titration to ensure it had not declined by more than 20%.

The experiment lasted 7 days (162 h), after which the overlying water was removed and a 5 cm diameter sub-
core was taken from each core. This was sectioned at 1 cm intervals and frozen. The remaining sediment was
sectioned at intervals of 0-1, 1-2, 2-3 and 3-5 cm, and sieved on a 500 μm mesh. Sediment and fauna remaining
on the sieve was preserved in buffered 10% formaldehyde in seawater. Fauna were picked from sieve residues
under a microscope, identified, and placed in glass vials or pre-weighed silver capsules.

### 2.3 Analysis

### 2.3.1 Bulk stable isotope analyses

Fauna samples were oven-dried at 45°C. Fauna with calcite skeletons (ophiuroids, molluscs and foraminifera)
were de-carbonated by the addition of a few drops of 6 N HCl. For soft-bodied fauna, 1 N HCl was used to
eliminate possible traces of carbonates. In all cases whole organisms were analysed. In the Loch Etive
experiment fauna from two replicate cores were allowed time to void their guts, but it was not clear that they
actually did so (see below). All samples were dried at ~50°C before analysis for OC content and $\delta^{13}$C.

Loch Etive samples were analysed on a Europa Scientific (Crew, UK) Tracermass isotope ratio mass
spectrometer (IRMS) with a Roboprep Dumas combustion sample converter. Appropriately sized samples of
acetanilide were used for quantification, and all C abundance data were blank corrected. Replicate analyses
revealed relative standard deviations of 4.6 % for C abundance and 0.7 ‰ for $\delta^{13}$C. Ythan sand flat samples were
analysed using a Flash EA 1112 Series Elemental Analyser connected via a Conflo III to a Delta$^{Plus}$ XP isotope
ratio mass spectrometer (all ThermoFinnigan, Bremen, Germany). Carbon contents of the samples were
calculated from the area output of the mass spectrometer calibrated against National Institute of Standards and
Technology standard reference material 1547 (peach leaves), which was analysed with every batch of ten
samples. The isotope ratios were traceable to International Atomic Energy Agency reference materials USGS40





and USGS41 (both L-glutamic acid); certified for $\delta^{13}C$ (‰). Long-term precisions for a quality control standard
(milled flour) were: total carbon $40.3 \pm 0.35$ %, and $\delta^{13}C$ -25.4 $\pm$ 0.13 ‰.
Overlying water samples were analysed for concentration and $\delta^{13}C$ of DIC as described by Moodley et al.
(2000). Briefly, a He headspace was created in sample vials, the $CO_2$ and $\delta^{13}C$ of which were quantified using a
Carlo Erba MEGA 540 gas chromatograph, and a Finnigan Delta S isotope ratio mass spectrometer,
respectively. The system was calibrated with acetanilide (Schimmelmann et al., 2009) and the IAEA-CH-6
standard. Repeat analyses of standard materials gave a relative standard deviation of 4.4% for DIC
concentrations, and a standard deviation of ±0.09‰ for $\delta^{13}C$.
**2.3.2 Bacterial phospholipid fatty acids(PLFA)**
Aliquots of sediment were treated with a Bligh and Dyer extraction, involving shaking at room temperature in a
2:1:1 mix of methanol, chloroform and water. Lipids were recovered in the chloroform layer, and were loaded
onto silica gel columns. Polar lipids were eluted in methanol, and methylated in the presence of methanolic
NaOH. The C12:0 and C19:0 fatty acid methyl esters were used as internal standards. Fatty acids were separated
by gas chromatography on a 30 m, 0.25mm i.d., 25 μm film thickness BPX70 column and combusted in a
Thermo GC-combustion II interface. Isotope ratios were then determined using a Thermo Delta+ isotope ratio
mass spectrometer (for further details see Woulds et al., 2014).
**2.4 Data treatment**
Uptake of added C by fauna is reported in absolute terms (see below), and as isotopic enrichments over the
natural background faunal isotopic composition. Isotopic compositions were expressed as $\delta^{13}C$, derived using
Eq. (1).

$$\delta\text{‰} = \left(\frac{Rs}{Rr} - 1\right) x\ 1000$$

247 (1)

Where $R_s$ and $R_r$ are the $^{13}C/^{12}C$ ratio in the sample and the reference standard respectively. Isotopic
enrichments ($\Delta\delta$) were then calculated using Eq. (2).

$$\Delta\delta = \delta^{13}C\ sample - \delta^{13}C background$$

250 (2)

Carbon uptake by faunal groups was calculated by subtracting naturally occurring $^{13}C$, multiplying by the
sample C contents, and correcting for the fact that the added phytodetritus was not 100 % $^{13}C$ labelled, as shown
in Eq. (3):

$$C\ Uptake_{sample} = \frac{\left(At\ \%_{sample} - At\ \%_{background}\right) X\ C\ Contents_{sample}}{At\ \%_{phytodetritus}}\ X100$$

254 (3)



Where At % is the $^{13}$C atoms present as a percentage of the total C atoms present. Data from individual
specimens was summed to produce faunal C uptake by different groups of fauna. For Loch Etive, background
$^{13}$C was subtracted based on natural faunal isotopic data collected concurrently with the C tracing experiment.
For the Ythan sand flat natural faunal isotopic data were not available, and instead the natural C isotopic
signature of sedimentary organic C (-20.2 ‰) was used. Isotopic signatures of fauna at the end of the
experiment had a maximum of 2460‰ and a mean of 175‰. Therefore the small inaccuracies introduced by the
use of this natural background value will not have been significant.
The DIC concentrations and δ$^{13}$C-DIC were used to calculate the total amount of added $^{13}$C present as DIC in
experimental chambers at each sampling time. A linear regression was applied to these to yield a separate
respiration rate for each core and for each period of respiration measurement (mean $R^2$ = 0.909, with the
exception of one measurement showing poor linearity with $R^2$ = 0.368), and the rate was multiplied by
experiment duration to calculate total respiration of added C during the experiment. In the case of the Ythan
sand flat respiration was measured during two separate 24 h periods through the experiment. In this case average
rates from the two measurements were used to calculate total respiration of added C throughout the experiment.
Bacterial C uptake was quantified using the compounds iC14:0, iC15:0, aiC15:0 and iC16:0  as bacterial
markers. Bacterial uptake of added C was calculated from their concentrations and isotopic compositions
(corrected for natural $^{13}$C occurrence using data from unlabelled sediment), based on these compounds
representing 14% of total bacterial PLFAs, and bacterial PLFA comprising 5.6% of total bacterial biomass
(Boschker and Middelburg, 2002). In the case of Loch Etive, the sediments from which PLFAs were extracted
had previously been centrifuged for porewater extraction, which could have led to a slight reduction in the
bacterial biomass and C uptake measured.
**3. Results**
The mean recovery of added C from the bacterial, faunal and respired pools together was 30±6% and 31±10%
of that which was added for Loch Etive and the Ythan sand flat respectively. This is a good recovery rate
compared to other similar experiments (e.g. Woulds et al., 2007). Most of the remaining C was likely left in the
sediment as particulate organic C.
**3.1 Remineralisation**
The average respiration rate of the added OC was similar in Loch Etive and the Ythan sand flat, and reached
0.64±0.4 and 0.63±0.12 mg C m-2h-1, respectively. Thus, the total amount of added C that was respired at each
site (over 156 h in Loch Etive and 162 h on the Ythan sand flat) was 99.5±6.5 and 102.6±19.4 mg C m-2 for
Loch Etive and the Ythan sand flat, respectively (Fig. 2). In both experiments, respiration rates measured in the
first 48 h (1.41±0.14 and 0.74±0.02 mg C m-2h-1 for Etive and the Ythan sand flat, respectively) were higher
than those measured in the last 48 h of the experiment (0.31±0.04 and 0.52±0.22 mg C m-2h-1 for Etive and the
Ythan sand flat, respectively; this difference was significant only for Loch Etive, t-test, P<0.001).
**3.2 Faunal biomass and C uptake**



Macrofaunal biomass in the experimental cores was 4337±1202 mg C m$^{-2}$ in Loch Etive and 455±167 mg C m$^{-2}$
on the Ythan sand flat. Macrofaunal δ$^{13}$C signatures (for individual specimens) reached maximal values of 7647
‰ and 2460 ‰ in Loch Etive and on the Ythan sand flat, respectively. Total faunal C uptake was orders of
magnitude greater in Loch Etive (204±72 mg C m$^{-2}$) than on the Ythan sand flat (0.96±0.3 mg C m$^{-2}$) (Fig. 2).
This difference was driven partly by a difference in biomass, but fauna on the Ythan sand flat were also
comparatively less active, as reflected by biomass specific C uptake at the two sites (0.047±0.01 and
0.0022±0.0006 mg C uptake per mg C biomass for Loch Etive and the Ythan sand flat respectively).
In Loch Etive, both faunal biomass and carbon uptake were dominated by two ophuroids, *Amphiura fillaformis*
and *A. chiajei*, which contributed 75 % and 95 % to the total biomass and to faunal C uptake, respectively (Fig.
3). The molluscs and polychaetes contributed 11 % and 6 % to biomass, but only 1.6 % and 1 % to faunal C
uptake, respectively. Amongst the polychaetes, the *Flabelligeridae* and *Harmothoe* tended to show lower $^{13}$C
enrichment (i.e. a lower specific uptake of labelled C), while representatives of all other families (*Capitellidae*,
*Syllidae*, *Cirratulidae*, *Cossura* and *Terebellidae*) showed much higher levels of labelling.
On the Ythan sand flat, the macrofaunal community was dominated by oligochaetes and nematodes (Fig. 3). The
proportion of total faunal C uptake accounted for by oligochaetes (48%) approximately matched their
contribution to faunal biomass (51%). However, nematodes contributed slightly less towards total faunal uptake
(14%) than they did to total biomass (19%). Other minor groups included amphipods (0.3% of biomass),
polychaetes (2% of biomass) and gastropods (1.5% of biomass). Of these groups, the polychaetes and
gastropods made disproportionately large contributions to faunal C uptake, accounting for 10% and 18%
respectively (Fig. 3).
In the Loch Etive experiment, metazoan meiofaunal and foraminiferal data were also collected. Metazoan
meiofaunal and foraminiferal biomass in experimental cores were 47±14 mg C m$^{-2}$ and 343±625 mg C m$^{-2}$,
respectively. These two groups showed maximal Δδ$^{13}$C values of 1360 ‰ and 3313 ‰, respectively. Metazoan
meiofauna were not taxonomically sorted, but amongst the foraminifera the highest labelling was observed in
*Crithionina sp.*, while *Pelosina* did not show measurable label uptake. Compared to the macrofauna, meiofaunal
C uptake was minor, at 0.18±0.20 and 5.21±5.15 mg C m$^{-2}$ for metazoans and foraminifera, respectively (Fig.
2). Thus, metazoan meiofauna and foraminifera contributed 1 % and 7 % to the total faunal biomass, and 0.1 %
and 2.5 % to faunal C uptake, respectively.
**3.3 Bacterial biomass and C uptake**
Bacterial biomass in the surface 5 cm of sediment in Loch Etive was 5515±3121 mg C m$^{-2}$, and on the Ythan
sand flat was 7657±3315 mg C m$^{-2}$. The amount of added C incorporated into bacterial biomass was two orders
of magnitude greater on the Ythan sand flat (127±89 mg C m$^{-2}$) than in Loch Etive (1.80±1.66 mg C m$^{-2}$, Fig. 2).
In the majority of cores, >90% of bacterial uptake occurred in the top 3 cm of sediment. However in one core
from Loch Etive, 28% of bacterial uptake occurred between 3 and 6 cm depth. In comparison, 52% of the
bacterial biomass from the top 5 cm occurred shallower than 3 cm for Loch Etive, and this value was 66% on
the Ythan sand flat. Biomass specific uptake for the bacteria was two orders of magnitude greater on the Ythan
sand flat (0.016±0.004 mg C uptake per mg C biomass) than in Loch Etive (0.00023±.00013 mg C uptake per



mg C biomass). Thus it appears that the rapid uptake of added C by bacteria at the sandy site was primarily
driven by a more active bacterial community, rather than by a larger bacterial biomass.
**3.4 Biological carbon processing patterns**
The large differences in macrofaunal and bacterial C uptake rates between the two sites resulted in markedly
different biological C processing patterns (Fig. 2). In both cases, respiration was an important, but usually not
the dominant, fate of biologically processed C, accounting for 25-60 %. In the case of Loch Etive, the dominant
fate of biologically processed C was macrofaunal uptake (64±10 %), and this also resulted in a greater amount
of total biological C processing (Fig. 2) than on the Ythan sand flat. On the Ythan sand flat bacterial uptake
(48±18 %) was the dominant fate of biologically processed C. In Loch Etive, uptake of C by bacterial, metazoan
meiofaunal and foraminiferal communities made only minor contributions to total biological C processing (Fig.
2). On the Ythan sand flat, macrofaunal uptake made a relatively minor contribution (Fig.2). Unfortunately,
uptake by meiofaunal organisms could not be quantified at the latter site.
**4 Discussion**
**4.1 Experimental approach**
This study compares data from two experiments which, while following the same principle of sediment core
incubations under natural conditions, nevertheless had slightly different experimental setups. The temperature,
core size, stirring regime, light availability and C dose added all differed between the two study sites. The
differences in stirring regime, temperature, and light availability were enforced to properly replicate natural
conditions in each experiment, thus any contrasts between experiments caused by these conditions are simply
reflections of actual differences in functioning of the two benthic habitats. The difference in core diameters was
driven by the practicality of collecting undisturbed sediment cores from the two contrasting sediment types. The
difference in C dose added was minor (~25%) and also driven by practical constraints. Previous studies have
found little impact of such relatively minor differences in C dose (Woulds et al., 2009). In cases where the
amount of added C has been observed to control biological processing patterns and rates, the difference in C
dose has been much more pronounced (10-fold, Buhring et al., 2006 b). Thus, while experimental details varied
between Loch Etive and the Ythan sand flat, we are confident that direct comparisons between the results of the
two experiments are valid and ecologically meaningful.
Due to practical constraints, meiofauna were not included in the analysis of the Ythan sand flat experiment.
Previous studies have found that meiofauna consume disproportionate amounts of C relative to their biomass
(Evrard et al., 2010), and that nematodes (a major meiofaunal group) made a negligible contribution to C
cycling (Moens et al., 2007). We are unable to speculate how active the meiofauna were in C cycling with
respect to their biomass in the present study but, despite wide variations in the importance of meiofaunal uptake
for the immediate fate of deposited organic C (Nomaki et al., 2005; Sweetman et al., 2009; Evrard et al., 2010),
meiofaunal C uptake is usually similar to or less than macrofaunal C uptake (Nomaki et al., 2005; Evrard et al.,
2010). Thus, we consider it unlikely that the meiofaunal community was involved in C processing on the same
scale as observed for bacterial uptake and total respiration, and exclusion of meiofauna in the Ythan sand flat
experiment is unlikely to have markedly altered the overall pattern of biological C processing that we observed.



There was a difference in the sieve mesh sizes used to collect macrofauna in the two experiments (300 μm in
Loch Etive and 500 μm on the Ythan sand flat). The use of larger mesh sizes is more conventional and more
practical in coarser grained sediments, such as those of the Ythan sand flat. The larger mesh used on the Ythan
sand flat is likely to have reduced the macrofaunal biomass recovered, and thus the macrofaunal C uptake
measured. However, the effect is likely to have been insufficient to explain the striking differences in
macrofaunal C uptake and biomass specific uptake seen between the two sites.
Finally, the majority of fauna were too small for manual removal of gut contents prior to analysis, and were
therefore analysed with their gut contents in place. The exception to this was two (out of four) of the Loch Etive
replicate cores, the fauna from which were placed in clean water and allowed time to void their guts before
freezing. However, this did not produce a significant difference in the mcarofaunal$^{13}$C pool between those cores
and the other two in which fauna retained their gut contents (Mann-Whitney U, p =0.245). Some infauna
respond to starvation (which would have been simulated by being placed in water without sediment present) by
retaining their gut contents for days or weeks. Therefore it is possible that many organisms either voided their
guts incompletely, or not at all. It is also possible that the amount of added C residing in macrofaunal guts was
comparatively small as shown by Herman et al. (2000), and so its exclusion from the analysis of fauna from two
replicate cores did not produce a difference that was measurable above the comparatively large variation in
faunal C uptake between cores caused by faunal patchiness. Thus it should be noted that the values reported here
as faunal C uptake include both C residing in gut contents and that which was assimilated into faunal tissue.
### 4.2 Respiration rates
The respiration rates observed in Loch Etive and on the Ythan sand flat were very similar (Fig. 2). In one sense
this is unsurprising, as the two experiments were conducted at the same temperature, and similar C loadings
were applied. Temperature is known to be a major control on sediment respiration rates, and benthic respiration
has been shown to respond to temperature changes with a $Q_{10}$ of 2-3 (Kristensen 2000). Increased temperatures
accelerate the diffusion of reactants and metabolites through the sediment, and also increase microbial process
rates. Further, after manipulating the temperatures at which cores from both a deep-sea and an estuarine site
were incubated, Moodley et al. (2005) found similar respiration rates of added phytodetritus at similar
temperatures, despite differences in water depth and faunal community. Thus, they showed that temperature can
be the dominant control on sediment community respiration rate. Our finding of similar rates of respiration in
response to added phytodetritus in Loch Etive and on the Ythan sand flat, despite marked differences in factors
which can influence respiration rates such as macrofaunal biomass, organic C concentration, and solute
transport processes (Kristensen, 2000; Hubas et al., 2007; Huettel et al., 2014), supports the suggestion that
temperature is the dominant control. This is in line with findings of a much wider study of ecosystem respiration
rates, in which their dependence on temperature was found to be remarkably similar across both terrestrial and
aquatic habitats, despite marked contrasts in taxa, biomass, and abiotic factors (Yvon-Durochet et al., 2015).
### 4.3 Faunal uptake
In the case of Loch Etive, the macrofauna overwhelmingly dominated total faunal C uptake (accounting for 97
%), compared to metazoan meiofauna (0.1 %) and foraminifera (2.5 %, ). These contributions were broadly



similar to their contributions to total faunal biomass (92 %, 1 % and 7 % for macrofauna, metazoan meiofauna and foraminifera respectively). Thus, in line with previous findings (Middelburg et al., 2000; Woulds et al., 2007; Hunter et al., 2012b), the distribution of C uptake amongst macrofauna, metazoan meiofauna and foraminifera was largely determined by the relative biomass of each group. The dominance of faunal C uptake by macrofauna (as opposed to meiofauna and forminifera) has been observed previously. For example, in shorter experiments on the Porcupine Abyssal Plain (Witte et al., 2003b), in the deep Sognefjord (Witte et al., 2003 a) and at certain sites on the Pakistan margin (Woulds et al., 2007), macrofauna dominated faunal C uptake, and at an Antarctic site Moens et al. (2007) found that meiofaunal nematodes made a negligible contribution to C uptake. However, uptake into the macrofaunal pool can be most important during the initial response to an OC pulse, with bacterial uptake and respiration becoming more important over longer timescales (Moodley et al., 2002; Witte et al., 2003 b). Also in contrast to the findings above, metazoan meiofaunal and foraminiferal uptake have been shown to be more important pathways for C in some situations (e.g. Moodley et al., 2000). Where macrofauna are absent, or where conditions do not favour their functioning, smaller taxa can come to dominate C uptake, such as within the Arabian Sea oxygen minimum zone (Woulds et al., 2007). At other sites, meiofauna and foraminifera have been shown to take up more C than macrofauna without the presence of a stress factor such as low oxygen. This was the case at 2170 m water depth in the NE Atlantic, where foraminifera dominated the initial uptake of added $^{13}$C (Moodley et al., 2002), and also in Sagami Bay, where Nomaki et al. (2005) observed foraminifera to take up more C than metazoan fauna. At a sandy subtidal site in the Wadden Sea, meiofauna was found to consume more C than macrofauna, despite the former having a lower overall biomass (Evrard et al., 2010).

The marked uptake of C by macrofauna in Loch Etive was largely driven by two species of ophuroid, which also dominated the macrofaunal biomass (Fig. 3). However, the ophuroids accounted for a greater percentage of total macrofaunal C uptake than they accounted for macrofaunal biomass (Fig. 3),and thus were disproportionately responsible for the large amount of added C that was routed into macrofaunal biomass and gut contents. On the Ythan sand flat, the contribution to C uptake by the dominant oligochaetes was in line with and therefore presumably controlled by their contribution to the biomass (both ~50%, Fig. 3). However, the other faunal groups present contributed differently to biomass and C uptake. Nematodes were responsible for less C uptake than might be expected from their biomass, while the rarer polychaetes, amphipods and molluscs fed comparatively efficiently on the added OM. This is in line with previous studies in which certain polychaete families have been found to be selective or rapid feeders on fresh algal detritus (e.g. Woulds et al., 2007).

When C uptake is plotted against biomass for each faunal specimen analysed across both study sites, a positive correlation is apparent (Fig. 4). This correlation has been reported previously (Moodley et al., 2005; Woulds et al., 2007), and suggests that total faunal C uptake is largely driven by faunal biomass, despite the fact they are auto-correlations (uptake data are derived by multiplying C contents of a specimen by its isotopic signature). Within each site the distribution of C uptake amongst faunal groups was also dominantly driven by biomass. However, the lower faunal biomass on the Ythan sand flat does not necessarily fully explain the lower faunal C uptake observed there, as biomass specific C uptake was also considerably lower than in Loch Etive. Therefore, the inter-site difference in faunal C uptake requires an additional explanatory factor. We suggest that the low




OC standing stock in the permeable sediment of the Ythan sand flat, supports a lower biomass and also less
active faunal community.
The identity of fauna responsible for C uptake was in line with expectations from some previous studies, but
contrary to those arising from others, and the reasons for this variation within the literature are not clear.
Therefore, while overall faunal uptake is dictated by biomass, it remains challenging to predict which faunal
groups and taxa will dominate C uptake in a particular benthic setting. This appears to be determined by the
complex interplay of factors that determine benthic community composition, such as the nature and timing of
food supply (Witte et al., 2003 a, b), environmental stressors (Woulds et al., 2007), feeding strategies and
competition (Hunter et al., 2012b).
**4.4 Total biological C processing rates**
Of our two study sites, Loch Etive showed a greater amount of total biologically processed C (Fig. 2). As both
sites showed very similar respiration rates of added C, the difference in total biological C processing was driven
by greater faunal uptake in Loch Etive (Fig. 2). The greater faunal uptake in Loch Etive was a result of greater
faunal biomass, as shown by the relationship between biomass and C uptake for the specimens analysed in this
study (Fig. 4).
Such a relationship between biomass and total biological C processing is also shown by data gathered from
previously published isotope tracing experiment results, where biomass data are also available. Data from the
experiments shown in Table 1 shows a correlation between total biomass (faunal plus bacterial) and total
biological C processing rate (Pearson's correlation, r=0.499, p=0.002).
We therefore suggest that benthic community structure impacts the total C processing capacity of benthic
environments, through a relationship between macrofaunal biomass and total biological C processing rates.
**4.5 Biological C processing categories**
The distribution of biologically processed C between different C pools (biological C processing pattern, Fig. 2)
varied markedly between the two sites. While they both showed similar proportions of biologically processed C
having been subjected to respiration, the dominant fate of such C in Loch Etive was uptake by macrofauna,
while on the Ythan sand flat it was uptake by bacteria (Fig. 2).
A review of previous isotope tracing experiments proposed a categorisation of biological C processing patterns
(Woulds et al., 2009), which can be used as a framework to explain patterns observed in this study.
Loch Etive was expected to show biological C processing pattern in line with the category labelled 'active
faunal uptake'. In this category, biological C processing is dominated by respiration, but faunal uptake accounts
for 10-25 % (Woulds et al., 2009). This category is found in estuarine and nearshore sites which are warmer
than the deep sea, have slightly more abundant OM, and thus support higher biomass and more active faunal
communities. However, the biological C processing pattern actually observed in Loch Etive was most similar to
the category labelled 'macrofaunal uptake dominated' (Fig. 5). In this category, uptake of C by macrofauna
accounts for a greater proportion of biologically processed C than total community respiration (Woulds et al.,



2009). This is a comparatively unusual pattern, previously only observed in the lower transition zone of the
Arabian Sea oxygen minimum zone. It was hypothesised in that case that the occurrence of a macrofaunal
population capable of C uptake of such magnitude was due to the presence of particularly high OC
concentrations in the sediment, coupled with sufficient oxygen for larger organisms (as opposed to at lower
oxygen concentrations within the oxygen minimum zone). This explanation also applies to the site studied here
in Loch Etive, where the sediment OC concentration was nearly 5 %. In contrast to the Arabian Sea site
however, our Loch Etive site featured fully oxygenated bottom water. Thus, the occurrence of macrofaunal
uptake dominated biological C processing appears to be facilitated by high OC availability and the resultant
faunal community, rather than by low oxygen conditions. A further indication of the control by OC availability
on the relative importance of faunal C uptake is shown in isotope tracing experiments conducted at sites in Pearl
Harbour impacted by invasive mangroves (Sweetman et al., 2010). A control site was OC poor (0.5% wt % OC)
and correspondingly showed respiration dominated biological C processing (Fig. 6). In contrast, a nearby site
from which invasive mangroves had been removed showed active (macro)faunal uptake (Fig. 6), in line with
increased sediment OC content (3.1% wt % OC) and an elevated macrofaunal biomass. A third site at which the
invasive mangroves were still present however showed respiration dominated C processing despite very high
OC concentration (8.2% wt % OC). However, in that case the unusual properties of mangrove detritus (being
tannin rich, with fibrous root mats binding the sediment) made the sediment inhospitable for many macrofauna
taxa, therefore bacterial C uptake and respiration was favoured over macrofaunal uptake (Sweetman et al.,

492 2010).

We hypothesised that the Ythan sand flat would show a biological C processing pattern that did not fit with
those laid out by Woulds et al. (2009). This hypothesis was supported, as biological C processing on the Ythan
sand flat was dominated by bacterial C uptake (Fig. 2). There have been indications in previous isotope tracing
experiments in sandy sediments of the German Bight that bacterial C uptake may be particularly important in
sandy sediments (Buhring et al., 2006a). Thus we now combine the previous and current results and use them to
propose a new biological C processing category labelled 'bacterial uptake dominated' (Fig. 5). In the new
category, bacterial uptake, rather than respiration, is the dominant fate of biologically processed C, accounting
for ~35-70 %. Respiration remains important, accounting for 25-40% of biologically processed C, and faunal
uptake tends to account for~5-20 %.
The new category of biological C processing so far has only been observed in two experiments targeting sandy,
permeable sediments, and so the features of such sediments appear to favour bacterial C uptake over faunal
uptake and total community respiration. Advective porewater exchange in permeable sediments has been shown
to enhance the rates of microbial processes such as remineralisation and nitrification (Huettel et al., 2014)
through rapid supply of oxygen and flushing of respiratory metabolites. This is balanced by introduction of fresh
OC as algal cells are filtered out of advecting porewater (Ehrenhauss and Huettel, 2004) and thus both the
substrate and electron acceptors for bacterial respiration are supplied. This efficient introduction of fresh OC is
consistent with the fact that Loch Etive and the Ythan sand flat showed similar bacterial biomass despite
considerable difference in OC concentration, thus OC supply rather than standing stock appears to be important
in determining bacterial biomass and activity.



While permeable sediments generally have similar or lower bacterial abundances than muddy sediments, their
bacterial communities tend to be highly active, and it has been suggested that, because they are subjected to
rapidly changing biogeochemical conditions, they are poised to respond rapidly to OC input (Huettel et al.,
2014). Notably however, the rapid rates of bacterial activity observed in permeable sediments do not typically
lead to build-up of bacterial biomass (Huettel et al., 2014). In a year-round study of permeability on the Ythan
sand flat, Zetsche et al. (2011) observed clogging of pore spaces during the summer, which was then removed.
Therefore one possible explanation for the lack of accumulation of bacterial biomass in permeable sediments is
regular removal of bacterial biomass during sediment reworking.
It is worth noting that the domination of biological C processing by bacterial uptake, to the extent that it is equal
to or greater than total community respiration, implies a high value for bacterial growth efficiency (BGE). This
parameter is calculated as bacterial secondary production divided by the sum of bacterial secondary production
and bacterial respiration, and thus represents the proportion of assimilated C that is routed into anabolism rather
than catabolism. Bacterial respiration is challenging to quantify, and is not quantified during isotope tracing
experiments. However, it is likely that a large proportion of total community respiration is attributable to
bacteria (Schwinghamer et al., 1986; Hubas et al., 2006). Thus, for the sake of discussion, BGE has been
approximated for the Ythan sand flat experiments as bacterial C uptake divided by the sum of bacterial C uptake
and total community respiration, giving a mean value of $0.51\pm0.18$ (this will be a conservative estimate). This
value is indeed at the high end of the range of values (<0.05 to >0.5) reported in a review of growth efficiency
for planktonic bacteria (Del Giorgio and Cole, 1998), but is in line with the modelled value of >0.5 for the most
productive coastal and estuarine sites in that same review. Bacterial growth efficiency is widely variable, both
spatially and temporally, and the factors which control it are not well understood. In the case of several potential
controlling factors, such as temperature and inorganic nutrient limitation, evidence is conflicting. However both
the rate of supply of organic substrate and its composition (bioavailable energy) seem to be positively correlated
with BGE. In particular increased supply of amino acids tends to increase BGE, and, amongst broad types of
OC, only that excreted by phytoplankton showed a high (>50%) mean BGE (Del Giorgio and Cole, 1998)
Bacterial growth efficiency also tends to increase from oligotrophic to eutrophic environments, and thus it often
correlates with primary productivity (Del Giorgio and Cole, 1998). These trends mean it is perhaps relatively
unsurprising that permeable sediments, with their potentially high input of fresh OC through filtering during
advective porewater flow have high BGE (Ehrenhauss and Huettel, 2004). In addition, it may be that BGE is
maximised if there is a shift in the relative proportions of bacterial cells that are highly active, versus those
which are dormant, inactive or dead (Del Giorgio and Cole, 1998). Furthermore, the proportion of highly active
cells has been found to increase with productivity. Thus, the high BGE observed on the Ythan sand flat (and in
the German Bight by Buhring et al., 2006) may be due to the fact that bacterial communities in permeable
sediments tend to be particularly active compared to those in cohesive sediments (Huettel et al., 2014).
Finally, faunal uptake was relatively minor in the Ythan sand flat experiment, and this suggests that bacterial C
uptake may have been favoured by a lack of competition from or grazing by macrofauna. A negative
relationship has previously been observed between macrofaunal biomass and bacterial C and N uptake in the
Arabian Sea, and a similar effect has been observed in the Whittard canyon (Hunter et al., 2012; 2013).



The biological C processing patterns presented in Fig.5 can accommodate most observations in the literature, but some findings do not fit in this conceptual scheme. For example, an experiment conducted in permeable sediments of the Gulf of Gdansk does not show the bacterial dominated biological C processing pattern that might be expected based on permeable sediment from the Ythan sand flat and the German Bight. Instead it shows respiration dominated biological C processing, with bacterial uptake, although greater than faunal uptake, responsible for only 16% (Fig. 6). Further, an OC rich site with invasive mangroves in Hawaii shows respiration dominated biological C processing, instead of the expected 'active faunal uptake' pattern (Fig. 6, Sweetman et al., 2010), however in this case the impact of mangrove roots on the sediment make it inhospitable to macrofauna.

Finally, bacterial uptake dominated biological C processing has also been observed over 3 days in sediments from the Faero-Shetland channel at a depth of 1080 m (Gontikaki et al., 2011). This is considerably deeper than all other observations, and the sediments in question contained a muddy fraction, although also featured grains up to gravel size. Thus this site does not fit the same general description as others showing bacterial uptake dominated biological C processing. In this case bacterial uptake dominated C processing was observed over the initial 3 days of the experiment, and after 6 days biological C processing was respiration dominated, more in line with expectations for the site. The authors explained the initial rapid uptake of C by bacteria as a reaction to the initially available reactive fraction of the added OM, before hydrolysis of the remaining OC began in earnest (Gontikaki et al., 2011).

In summary, the proposed categorisation of biological C processing patterns works well across many different sites, but variation in characteristics of individual sites can still lead to some unexpected results.

## 5 Conclusions

The rate of respiration of added phytodetritus was dominantly controlled by temperature, rather than other factors such as benthic community biomass, sediment OC concentration, or solute transport mechanism.

Faunal C uptake was related to faunal biomass. Further, total biological C processing rates in this and previous studies appear to be dominantly determined by benthic biomass. Therefore benthic community structure has a role in controlling the C processing capacity of benthic environments.

A new biological C processing pattern category was proposed titled 'bacterial uptake dominated', which seems usually to be observed in permeable sediments, where conditions are particularly conducive to active bacterial populations.



**Author contributions**


C. Woulds designed and conducted the experiments with input from G. Cowie, J. Middelburg and U. Witte.
Sample analysis was completed by C. Woulds, S. Bouillon and E. Drake. C. Woulds prepared the manuscript
with the assistance of all co-authors.

**Acknowledgements**


The authors would like to thank Eva-Maria Zetsche, Val Johnson, Owen McPherson, Caroline Gill and Gwylim
Lynn for assistance with the Ythan sand flat fieldwork, and Matthew Schwartz, Rachel Jeffreys, Kate Larkin,
Andy Gooday and Christine Whitcraft for assistance with the Loch Etive fieldwork. Jonathan Carrivick created
Figure 1. The work was funded by the Natural Environment Research Council and the Netherlands Earth
System Science Center.




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




| Source | Site/Experiment | Depth (m) | Temperature (°C) | Macrofaunal Biomass (mg C m$^{-2}$) | Bacterial Biomass (mg C m$^{-2}$) | Respiration Rate (mg C m$^{-2}$ h$^{-1}$) | Total Processing Rate (mg C m$^{-2}$ h$^{-1}$) |
|---|---|---|---|---|---|---|---|
| Moodley et al. 2000 | Oosterschelde | Intertidal | 10 | nd | nd | 7.758 | 13.150 |
| Moodley et al. 2002 | NW Spain | 2170 | 3.6 | 39 | 2 | 0.083 | 0.290 |
| Witte et al. 2003 b | PAP 60h | 4800 | nd | 120 | 2500 | 0.167 | 0.225 |
| Witte et al. 2003 b | PAP 192h | 4800 | nd | 120 | 2500 | 0.167 | 0.188 |
| Witte et al. 2003 b | PAP 552h | 4800 | nd | 120 | 2500 | 0.236 | 0.263 |
| Witte et al. 2003 a | Sognefjord 36h | 1265 | 7 | 250 | 8500 | 0.539 | 0.781 |
| Witte et al. 2003 a | Sognefjord 72h | 1265 | 7 | 250 | 8500 | 0.451 | 0.715 |
| Moodley et al. 2005 | N. Sea (perturbed) | 37 | 6 | 756 | 2688 | 0.600 | 0.735 |
| Moodley et al. 2005 | N. Agean | 102 | 14 | 73 | 522 | 2.895 | 3.075 |
| Moodley et al. 2005 | N. Agean | 698 | 14 | 37 | 366 | 3.110 | 3.290 |
| Moodley et al. 2005 | E. Med. | 1552 | 14 | 6 | 254 | 2.750 | 2.830 |
| Moodley et al. 2005 | E. Med. | 3859 | 14 | 4 | 312 | 2.495 | 2.610 |
| Moodley et al. 2005 | NE Atlantic 24h | 2170 | 4 | 138 | 313 | 0.300 | 0.330 |
| Moodley et al. 2005 | N. Sea | 37 | 16 | 732 | 2304 | 3.025 | 3.600 |
| Moodley et al. 2005 | Estuary | Intertidal | 18 | 1356 | 1260 | 2.545 | 3.705 |
| Bhuring et al. 2006 | German Bight 12h | 19 | 9 | nd | nd | 0.258 | 3.592 |
| Bhuring et al. 2006 | German Bight 30h | 19 | 9 | nd | nd | 0.620 | 2.523 |
| Bhuring et al. 2006 | German Bight 132h | 19 | 9 | nd | nd | 0.258 | 0.667 |
| Bhuring et al. 2006 | German Bight in situ | 19 | 13 | nd | nd | 0.338 | 2.834 |
| Woulds et al. 2009 | PM pre 140 2d | 140 | 22 | 110 | 1100 | 2.827 | 3.750 |
| Woulds et al. 2009 | PM post 140 2d | 140 | 22 | 110 | 1100 | 2.066 | 2.977 |
| Woulds et al. 2009 | PM post 140 5d | 140 | 22 | 110 | 1100 | 1.164 | 1.611 |
| Woulds et al. 2009 | PM post 140 in situ | 140 | 22 | 110 | 1100 | 0.705 | 0.955 |
| Woulds et al. 2009 | PM pre 300 2d | 300 | 15 | 0 | 1000 | 0.365 | 0.487 |
| Woulds et al. 2009 | PM pre 300 5d | 300 | 15 | 0 | 1000 | 0.285 | 0.386 |
| Woulds et al. 2009 | PM post 300 2d | 300 | 15 | 0 | 1000 | 0.527 | 0.931 |
| Woulds et al. 2009 | PM post 300 5d | 300 | 15 | 0 | 1000 | 0.477 | 0.865 |
| Woulds et al. 2009 | PM post 300 in situ | 300 | 15 | 0 | 1000 | 0.035 | 0.250 |
| Woulds et al. 2009 | PM pre 850 2d | 850 | 10 | nd | nd | 1.064 | 1.934 |
| Woulds et al. 2009 | PM pre 940 5d | 940 | 9 | 910 | 700 | 0.469 | 0.933 |
| Woulds et al. 2009 | PM post 940 5d | 940 | 9 | 910 | 700 | 0.486 | 1.274 |
| Woulds et al. 2009 | PM post 940 in situ | 940 | 9 | 910 | 700 | 0.155 | 0.986 |
| Woulds et al. 2009 | PM pre 1000 2d | 1000 | 8 | nd | nd | 0.990 | 2.411 |
| Woulds et al. 2009 | PM pre 1200 5d | 1200 | 7 | 60 | nd | 0.274 | 0.289 |
| Woulds et al. 2009 | PM pre 1850 2d | 1850 | 3 | 110 | 300 | 0.065 | 0.235 |
| Woulds et al. 2009 | PM pre 1850 5d | 1850 | 3 | 110 | 300 | 0.434 | 0.506 |
| Woulds et al. 2009 | PM post 1850 5d | 1850 | 3 | 110 | 300 | 2.459 | 2.623 |
| Sweetman et al. 2010 | Pearl Harbour Control | Intertidal | 24 | 337 | 5500 | 3.835 | 4.343 |
| Sweetman et al. 2010 | Pearl Harbour Removal | Intertidal | 24 | 3391 | 4500 | 5.349 | 7.401 |
| Sweetman et al. 2010 | Pearl Harbour Mangrove | Intertidal | 24 | 713 | 18154 | 5.456 | 6.048 |
| Sweetman et al. 2010 | Kaneohe Bay Control | Intertidal | 24 | 882 | 3500 | 6.125 | 6.849 |
| Sweetman et al. 2010 | Kaneohe Bay Removal | Intertidal | 24 | 1435 | 9000 | 5.295 | 7.475 |
| Evrard et al. 2010 | Wadden Sea | Photic Subtidal | 15 | nd | nd | 0.031 | 0.034 |
| Evrard et al. 2012 | Gulf of Gdansk (sandy) | 1.5 | 20 | 558 | 407 | 0.047 | 0.061 |
| This study | Loch Etive | 70 | 11 | 4337 | 5515 | 0.638 | 1.994 |
| This study | Ythan sand flat | Intertidal | 11 | 455 | 7657 | 0.633 | 1.421 |


Table 1. Sources and site details of previous isotope tracing experiment data. PAP = Porcupine Abyssal Plain.
For Woulds et al. (2009) experiments PM = Pakistan Margin, 'pre' and 'post' indicate pre- or post-monsoon
seasons, and 2d or 5d indicate approximate experiment durations in days. In some other cases experiment
durations are indicated in hours (h).





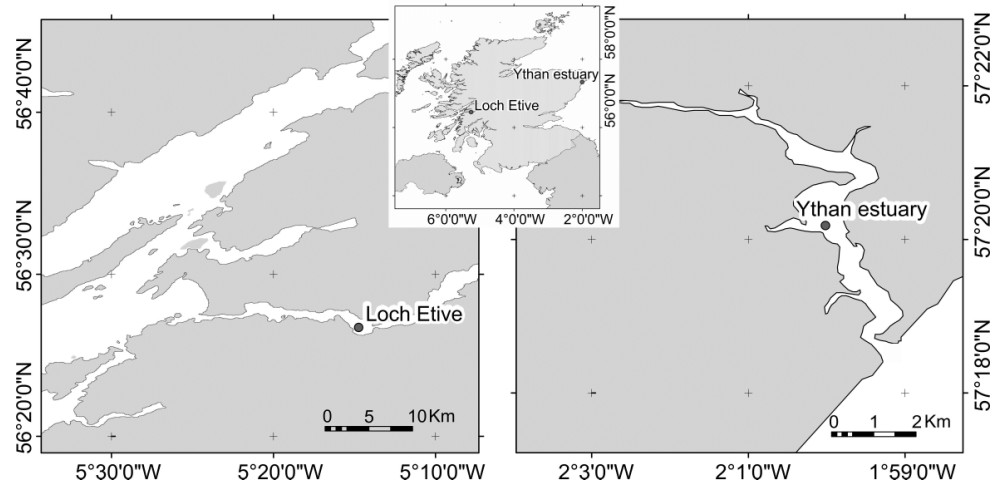


Figure 1. Map showing site locations.





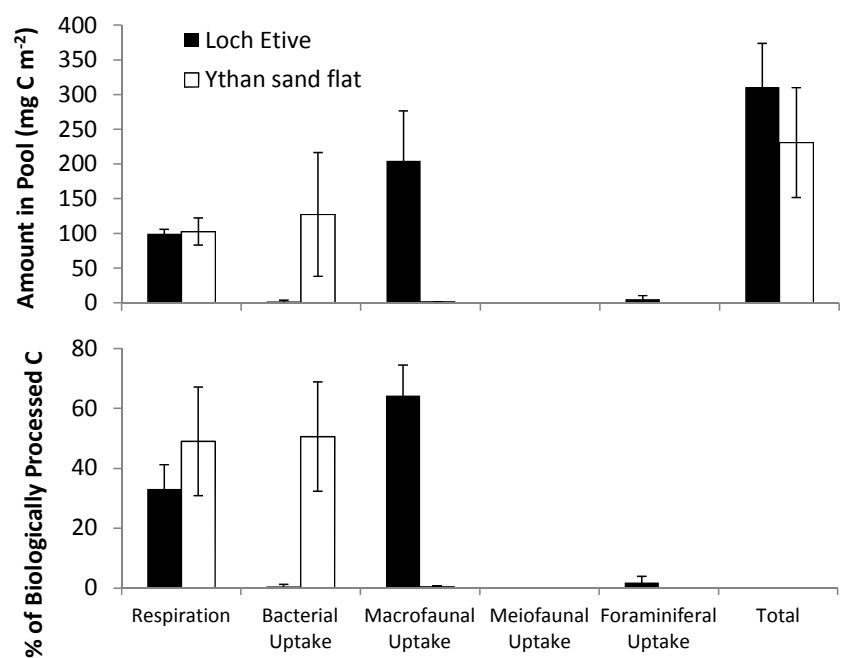


Figure 2.The distribution of initially added C between different biological pools at the end of the experiments in
absolute terms (upper panel), and as percentages of total biological C processing (lower panel).





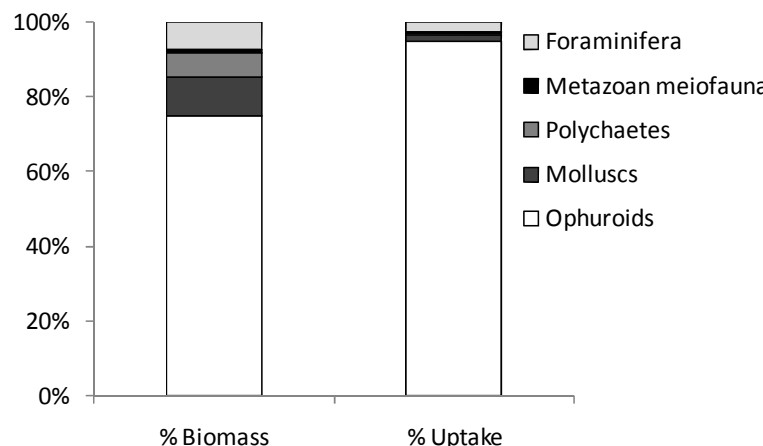


A

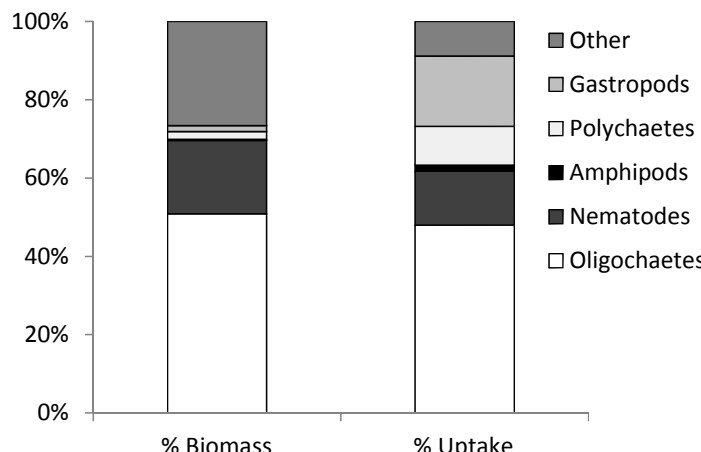


B
Figure 3.Taxa responsible for biomass and C uptake in a) Loch Etive, and b) the Ythan sand flat.




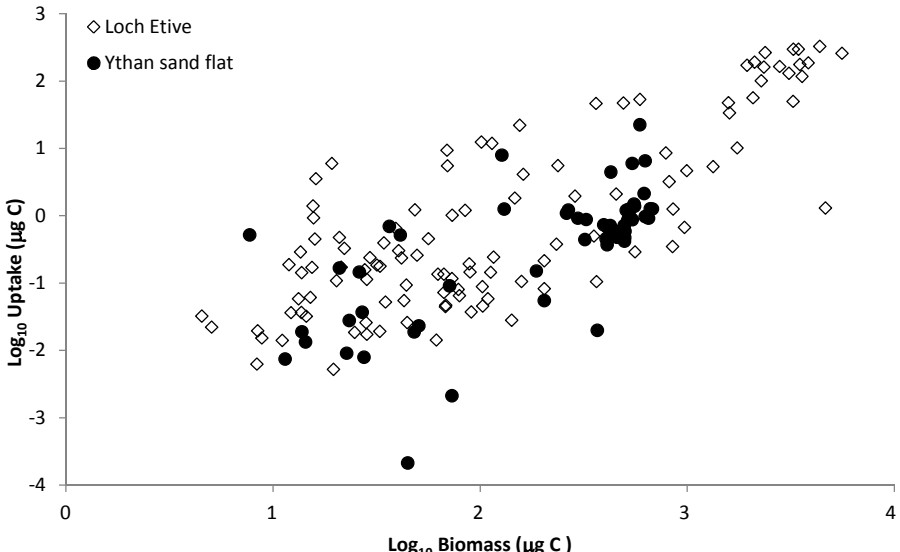


Figure 4. Log$_{10}$ uptake against Log$_{10}$ C biomass for all specimens analysed in Loch Etive and on the Ythan sand

flat.




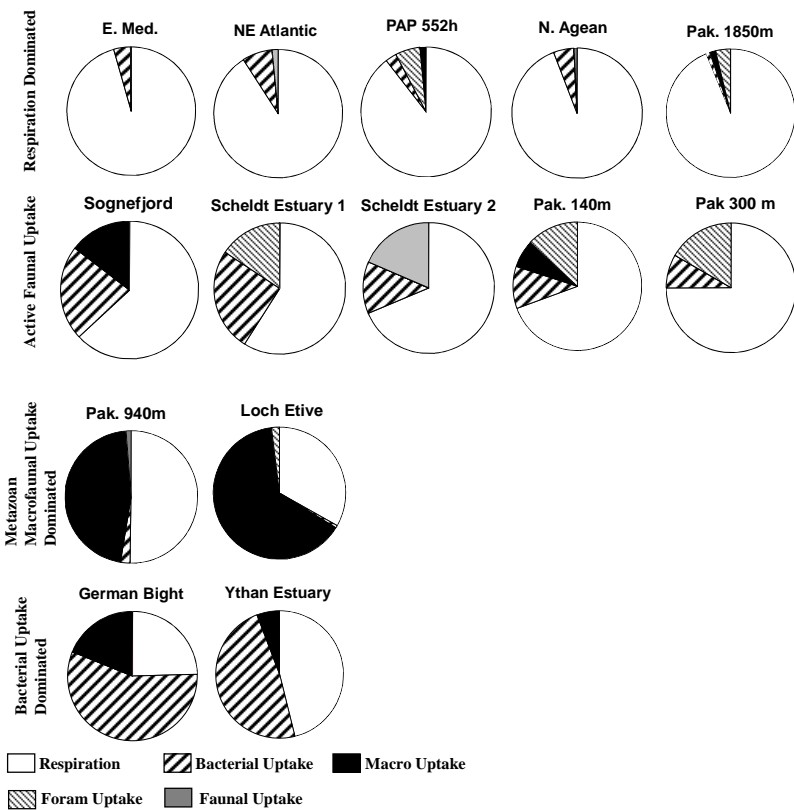


Figure 5. Biological C processing pattern categories adapted from Woulds et al. (2009), with the experiments

from this study and the new category 'bacterial uptake dominated' added. Data sources are as follows; Eastern

Mediterranean (E. Med.), NE Atlantic, North Aegean (N. Aegean) and Scheldt Estuary 2: Moodley et al. (2005);

Porcupine Abyssal Plain (PAP 552 h): Witte et al. (2003 b); Pakistan Margin (Pak. 140 m, 300 m, 940 m, 1850

m): Woulds et al. (2009); Sognefjord: Witte et al. (2003 a); Scheldt Estuary 1: Moodley et al. (2000); German

Bight: Buhring et al., (2006).




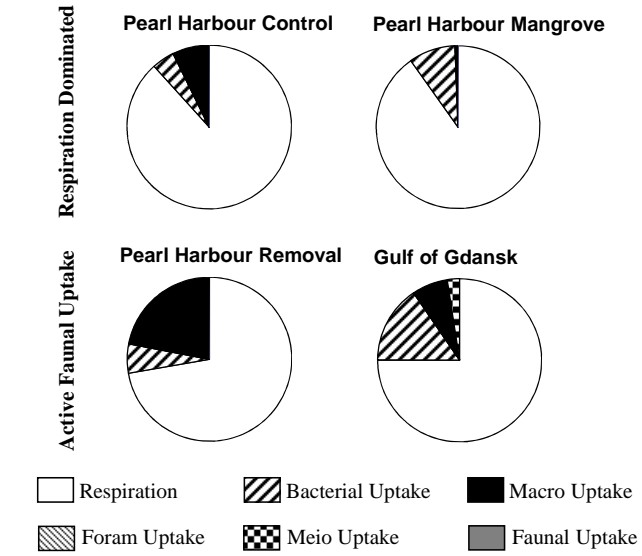


Figure 6.Biological C processing categories in two recent studies. Pearl Harbour data are from Sweetman et al.
(2010), Gulf of Gdansk data are from Evrard et al. (2012).