# Peer review of "Patterns of carbon processing at the seafloor: the role of faunal"

_Biogeosciences, 2016_

## Referee Comment (RC1) · Anonymous Referee #1 · 11 Mar 2016

This paper described carbon processing at two different sites (fine-grained, high TOC concentration site and coarse-grained, low TOC site) at shallow water depths. Although the experiments were not totally comparable for some reasons (see details below), the experimental results are clear and meaningful; comparable respiration rates at both sites, but high macrofaunal uptake at the high TOC site while high bacterial uptake at the low TOC site. However, I think the discussion regarding carbon processing categorization has critical problems and needed to be removed, or presented after considerable, extensive modification.

Major comments:

1. Carbon processing categorization The discussion 4.5. based on many uncertainty

and speculations, and need to remove from the manuscript. The authors proposed the categorization of C processing using data in this study and references. However, there is no mention on how and why authors selected specific time scale of the incubation duration. In Woulds et al. (2009), there were circle graphs of carbon fate for both ∼2 days and ∼5 days. However, in this paper, only one of them (I guess so) are shown. It is expected that the respired C increases with time (as mentioned in the line 563) while macrofaunal and bacterial 13C-label will be respired and decreased. Further, the faunal uptake and bacterial uptake also showed different patterns with time between taxa: for instance, macrofauna responded quicker than foraminifera (Witte et al. 2003, Nature), bacterial assimilation decreased after 1 or 2 days (Middelburg et al. 2000) whereas foraminiferal uptake showed increasing pattern during similar time scale (Moodley et al. 2000). It is thus obvious that the time scale selection is the most important factor to properly categorize the carbon processing. In this manuscript, data from different time scales (hours to 23 days) were combined without description what time scale of incubation was selected in the categorization from several different incubation periods (e.g. Moodley et al. 2002, Witte et al. 2003a, b, Bhuring et al. 2006). Also, there is no discussion on the effect of time scale (except line 563, which mentioned as to explain the irregular pattern of the categorization). I therefore recommend to remove discussion 4.5 from the manuscript and just discuss Loch Etive was macrofauna dominated C processing and Ythan sand flat was bacteria dominated. The manuscript itself can withstand as research paper without the chapter 4.5.

2. Differences in light condition. The authors performed the 13C-labeled phytodetritus experiments with and without light (with light: Loch Etive, without light: Ythan sand flat). The authors validate the different conditions because natural environments are dark and light conditions, respectively. However, I believe that the incubation with light makes complicated pathways. Without light, the 13C-phytodetritus is incorporated into heterotrophic microbes or eukaryotes, and either assimilated into their biomass or respired as 13CO2. With light, however, the respired 13CO2 can be assimilated into photoautotrophic microbial biomass via photosynthesis. This leads underestimation of respired carbon and overestimation of bacterial assimilation. Without light, chemolithoautotrophic microbes can also cause same process, but the contribution must be smaller than photosynthesis. How much proportion of $CO_2$ was labeled with 13C? If the 13C concentrations in $CO_2$ is almost negligible (few %), then the bacterial assimilation via photosynthesis may also be negligible. This can be calculated from the DIC-d13C data of the study. Or, if there are literature which investigated bacterial community at this area, then the authors may validate that photoautotrophic bacteria was minor.

3. Uptake calculation The authors calculated the Carbon uptake by sample with the equation (3), line 253. However, the At% phytodetritus must be subtracted by At% background. I understand that the extent of 13C-label in this study (25% and 34%) are high and the re-calculated values using subtracted value may change only 2 or 3 % (considering 25 become 23.9 and 34 become 32.9). However, the it is necessary to indicate appropriate values as much as possible.

Other specific comments. Line 32 Did the accessibility by bacteria to added C similar between two sites? Please show the vertical profiles of 13C if possible.

Line 145. Figure 1 does not show any sills or geographical names. Please include these information to the figure or delete the citation (Fig.1 ) from the end of this sentence.

Line 163. While the Loch Evive site has 70 m water depth, the Ythan estuary site exposed during low tide. This is a great difference between two sites, in addition to sediment grain size and OC concentrations. The authors need to discuss the potential impacts of these differences of OC cycling and validate why the authors did not perform the experiment at coarse grained, OC poor site having similar water depths (or vice versa).

Line 171. What exactly was the phytodetritus labeled with 13C? Was that degraded in some way? Or some sort of algal species? Was this same to the one which was added

to Ythan sand flat? Please clarify these details.

Line 173. How much volume was the overlying water in the core?

Line 185 150 um sieve is not typical size separation for meiofauna. Why did the authors choose this size?

Line 189 Why the authors used milliQ water instead filtered seawater of artificial seawater? MilliQ water may had elution of organic matters from fauna due to osmotic shock (although the results showed insignificant effect).

Line 196 Bubbling with air in this experiment while the Loch Etive site cores were maintained with oxystat system. How did this affect to $^{13}C$-$CO_2$ amounts?

Line 253 The equation is not presented in correct way (no under bar below "C Uptake sample". What the unit of "C Uptake sample"?

Line 263 It is not clear about the linear regression. Do the authors mean linear regression of different incubation periods? It is also important to show the changes in $d^{13}C$-DIC (or $^{13}C$-respiration rates) with time, because the changes in $^{13}C$-respiration with time should give crucial info regarding faunal or bacterial responses and C processing.

Line 267 It is necessary to show the respiration data of Ythan sand flat, too, as Table or supplementary figure.

Line 274 Please describe the centrifuge condition ( x g, how long, and what temperature etc). It will help to guess the potential effects of centrifuge on bacterial PLFA loss.

Line 279. Did the authors examine the $d^{13}C$ of bulk sediments? If so, please include as Table etc.

Line 282 Again, it is important to show temporal changes in $d^{13}C$ (or respired $^{13}C$).

Line 326. 0.00023 mgC per mgC corresponds ~5 or 10 per mil of Dd13C, which is relatively low labeling. What were the variation in d13C of natural PLFA and labeled PLFA? Can you add as Table?

Lines 347 to 353. Whatever the C dose amounts were similar, the authors should think about the difference in natural phytodetritus supply rates at two sites. The same amount of 13C-phytodetritus input should have completely different effects on between originally eutrophic (in terms of OM) site and oligotrophic site. The authors should discuss these point of view by referring the primary production rates at two sites.

Line 368 Can you cite any paper which dealing different size screens?

Lines 376 to 380. Due to the osmotic shock by milliQ water (according to M&M), the fauna may be dead and did not have a time to void the gut.

Line 431. Gooday et al. 2008 represent biomass-uptake relationships with different symbols for bacteria, fauna, foraminifera. Can you also make such kind of Figure 4 for better comparison?

Line 438. This may suggest that the macrofauna of Ythan sand flat has low background metabolism than Loch Etive.

Line 459. I cannot follow why the authors said "macrofaunal biomass" in this sentence whereas the line 456 mentioned "biomass (faunal plus bacterial)". Please describe more in detail if the authors actually intended to say "macrofaunal biomass".

Chapter 4.4. can be combined to 4.3.

Line 520. Both methods (Total respiration rate measurements and bacterial C assimilation rates) has considerable uncertainty. Thus the discussion here, dealing bacterial growth efficiency, is somewhat over-interpretation. Also, as mentioned earlier, because the incubation of Ythan sand flat sediment was carried out under light condition, it is possible that some 13C-bacterial lipids were originated from the photoautotrophic microbes, not by heterotrophic bacteria which incorporated 13C-labeled phytoplankton.

Line 571 Again, temporal changes in DIC-13C at both site may give better idea about these interpretations.

Line 673 Hunter et al. 2012b. There is no Hunter et al. 2012a, thus deleted "b".

Table 1 Please add a new column showing incubation periods.

Figure 2. Please add "n.d." for meiofauna and foraminifera of Ythan sand flat.

---

## Referee Comment (RC2) · Anonymous Referee #2 · 16 Mar 2016

Woulds et al. present data and discussion on two pulse-chase 13C-labeled organic matter experiments conducted on sediments cores taken from two estuarine sediment types in Scotland. The goal of the experiments was to investigate the flow of freshly deposited organic matter through the benthic food web. These experiments follow in the footsteps of a limited number of similar pulse-chase experiments that have been conducted thoughout the last decade or so. It should be clear that the two experiments described here in this study, while similar in nature, are not directly comparable (e.g. differing coring diameters, different seasons, etc..). Nevertheless, the comparison of two types of estuarine benthic environments yields insights into the "processing" of freshly deposited organic matter.

[Figure]

A major strength of this manuscript is the comparison made between the sites investigated in this study and previous studies of "biological C processing". As the authors point out, there are all sorts of caveats to be attached to such experiments, and reproducibility is certainly bound to be one of the problems. Only through repeated experimentation with a broadly similar approach does a general picture begin to emerge. Overall, this manuscript represents a good contribution to the steadily expanding range of data and results from such experiments.

The first half of the manuscript through the Results section is generally well-written. Some issues need to be addressed, and these are noted below. In my opinion, however, the Discussion could be easily halved. For instance, the discussion of BGE (lines 520 to 545) far exceeds the data supporting any conclusions. Or see my comments below on Section 4.2 and the paragraph starting at Line 493. Overall, a more focused discussion would greatly benefit the overall impact of the manuscript.

Specific comments:

1. Line 73: It might be worth pointing out what does biological C processing not cover. Is there non-biological C processing in these systems? It might be worth pointing out the differences.

2. Line 76: A quibble: Stable isotope tracer experiments are an excellent tool, but not ideal. For instance, radiotracer 14C incubations are far more sensitive and do not depend on sorting out mass of naturally occurring background tracer distribution.

3. Line 117 and following: Independent of the food-web tracer studies, it would be nice to have some information on the relative benthic biomasses for these two sediment types, e.g. muddy and sandy bottoms. I would be surprised if muddy bottoms actually supported more faunal biomass.

4. With the exception of the respiration measurements, these are single endpoint experiments. Dynamics between the pools are not necessarily accessible.

5. Line 124: "Recent findings" is relative; dynamic biogeochemical cycling in low OC permeable sediments has been extensively documented over the last two decades.

6. Line 171: Please describe more carefully the labeled phytodetritus in more detail. Was it composed of a single species and what? Was it prepared in the same fashion for both sites? What was it composed of? How fresh was it? Was it added as fresh or freeze-dried material.

7. Does the difference between the labeling percentages (ca. 25% and 34%) for the two sites reflect different batch preparations, or differing compositions of phytodetritus?

8. Methods: It's not entirely clear to me that total bulk 13C of the sediment was determined (i.e. total Corg 13C). This must have been done in order to calculate the recoveries of tracers shown in Figure 2.

9. Is there a time zero sample, i.e. samples taken from one core immediately after the addition of the 13C-labeled phytodetritus?

10. Line 244 and following: It is not really clear to me why the authors work with the del ($\delta$) notation for these type of experiments. There is also no obvious connection from how they go from Equation 2 to Equation 3, the latter of which is the more relevant for this manuscript.

11. Calculations with exceedingly large enrichments, for instance as seen in the macrofaunal biomass (lines 290 and following), become inaccurate.

12. Line 280: ….or as dissolved organic carbon.

13. Section 3.1: It might be helpful for the reader to plot the remineralization data over the time course of the experiment.

14. Section 4.2: This whole discussion is rather contradictory. On one hand the authors claim that the temperature and organic C loading are similar (line 384), but then suggest that temperature plays a larger role than biomass or organic C (lare 395) does

not make sense. Furthermore, there are no proper controls for assessing any of these factors. I would drop this whole discussion (see further discussion about curtailing discussion) and the conclusions regarding temperature (line 571). This was not the point of the study, it was not properly assessed, nor is it supported by the data.

15. Line 494: "This hypothesis...." Which hypothesis? From this paper or Woulds et al. 2009? Actually I find the whole discussion of hypotheses, both here and earlier in the manuscript (line 134 and following) a bit specious. I think that it is enough for the authors to state that they are comparing two types of sites that are thus far lacking from the overall range of sites on which such experiments have been performed.

16. Figure 2: I assume that "Total" stand for the sum of respiration, bacterial uptake, etc.. in that case, there is also no Total for the % of Biologically Processed C. It might also be interesting to add a third panel to include total initial pool size of each of the separate pools (i.e. how much C is in each pool originally).

17. Figures 5 and 6: These figures are quite compelling, although I think that they could be combined. It is not entirely obvious why there are two separate figures.

―――――――――――――――――――――――

---

## Author Comment (AC2) · 5 Apr 2016

Reviewer 2:

Once again we would like to thank the reviewer for their overall positive opinion, and for their attention to detail which will allow us to improve the manuscript. Major comments:

The main comment from this reviewer is that the discussion is overly long. We agree, especially concerning the section about bacterial growth efficiency, and will undertake to significantly reduce the length of the discussion. Specific comments:

Reveiwer: Line 73: It might be worth pointing out what does biological C processing not cover. Is there non-biological C processing in these systems? It might be worth

pointing out the differences.

Answer: The term is used to distinguish between short term uptake and cycling and longer tern C burial. This will be clarified.

Reveiwer: Line 76: A quibble: Stable isotope tracer experiments are an excellent tool, but not ideal. For instance, radiotracer 14C incubations are far more sensitive and do not depend on sorting out mass of naturally occurring background tracer distribution.

Answer: Acknowledged, but working with stable isotopes has practical benefits which can allow increased numbers of experimental treatments/durations/replicates. A note will be added.

Reveiwer: Line 117 and following: Independent of the food-web tracer studies, it would be nice to have some information on the relative benthic biomasses for these two sediment types, e.g. muddy and sandy bottoms. I would be surprised if muddy bottoms actually supported more faunal biomass.

Answer: Detail will be added.

Reveiwer: With the exception of the respiration measurements, these are single end-point experiments. Dynamics between the pools are not necessarily accessible.

Answer: Agreed, wording will be amended.

Reveiwer: Line 124: "Recent findings" is relative; dynamic biogeochemical cycling in low OC permeable sediments has been extensively documented over the last two decades.

Answer: Agreed, wording will be amended.

Reveiwer: Line 171: Please describe more carefully the labeled phytodetritus in more detail. Was it composed of a single species and what? Was it prepared in the same fashion for both sites? What was it composed of? How fresh was it? Was it added as fresh or freeze-dried material.

Answer: Detail will be added.

Reveiwer: Does the difference between the labeling percentages (ca. 25% and 34%) for the two sites reflect different batch preparations, or differing compositions of pytodetritus?

Answer: Detail will be added as above.

Reveiwer: Methods: It's not entirely clear to me that total bulk 13C of the sediment was determined (i.e. total Corg 13C). This must have been done in order to calculate the recoveries of tracers shown in Figure 2.

Answer: The totals shown in figure 2 are total biologically processed C, and therefore do not contain C remaining in the sediment.

Reveiwer: Is there a time zero sample, i.e. samples taken from one core immediately after the addition of the 13C-labeled phytodetritus?

Answer: This is only available for Loch Etive, and not on the Ythan sand flat, therefore data has not been included.

Reveiwer: Line 244 and following: It is not really clear to me why the authors work with the del (_) notation for these type of experiments. There is also no obvious connection from how they go from Equation 2 to Equation 3, the latter of which is the more relevant for this manuscript.

Answer: Data are reported using the del notation in the results section because many workers in the field use this notation, and ïĄĎïĄď is a clear way of displaying isotopic enrichments. However, our calculations for uptake used At% instead. There is not supposed to be a connection between equation 2 and equation 3.

Reveiwer: Calculations with exceedingly large enrichments, for instance as seen in the macrofaunal biomass (lines 290 and following), become inaccurate.

Answer: The reviewer's meaning is not quite clear, however if this is given as a reason

for not using the del notation, then note that uptake calculations were made using At% instead.

Reveiwer: Line 280: . . ..or as dissolved organic carbon.

Answer: This will be added.

Reveiwer: Section 3.1: It might be helpful for the reader to plot the remineralization data over the time course of the experiment.

Answer: We will prepare these plots and assess whether they are a good use of space.
* * *

---

## Author Response (AR1)

**1 BG-2016-14 Response To Reviews**

**2 Reviewer 1 Major Comments:**

We would like to thank the reviewer for their thorough review, and for their overall positive opinion.

Reviewer: Carbon processing categorization The discussion 4.5. based on many uncertainty and 4 speculations, and need to remove from the manuscript. The authors proposed the categorization of 5 6 C processing using data in this study and references. However, there is no mention on how and why 7 authors selected specific time scale of the incubation duration. In Woulds et al. (2009), there were circle graphs of carbon fate for both \_2 days and \_5 days. However, in this paper, only one of them (I 8 9 guess so) are shown. It is expected that the respired C increases with time (as mentioned in the line 563) while macrofaunal and bacterial 13C-label will be respired and decreased. Further, the faunal 10 uptake and bacterial uptake also showed different patterns with time between taxa: for instance, 11 12 macrofauna responded quicker than foraminifera (Witte et al. 2003, Nature), bacterial assimilation 13 decreased after 1 or 2 days (Middelburg et al. 2000) whereas foraminiferal uptake showed 14 increasing pattern during similar time scale (Moodley et al. 2000). It is thus obvious that the time 15 scale selection is the most important factor to properly categorize the carbon processing. In this 16 manuscript, data from different time scales (hours to 23 days) were combined without description 17 what time scale of incubation was selected in the categorization from several different incubation periods (e.g. Moodley et al. 2002, Witte et al. 2003a, b, Bhuring et al. 2006). Also, there is no 18 discussion on the effect of time scale (except line 563, which mentioned as to explain the irregular 19 20 pattern of the categorization). I therefore recommend to remove discussion 4.5 from the manuscript 21 and just discuss Loch Etive was macrofauna dominated C processing and Ythan sand flat was bacteria 22 dominated. The manuscript itself can withstand as research paper without the chapter 4.5.

Answer: The reviewer is correct that in the medium and longer term the experiment duration will have an effect on biological C processing pattern, with respiration becoming more important with time (and in the end we might expect C which was incorporated into biomass to be respired as well, such is the nature of a pulse chase experiment). Our manuscript concerns the short-term biological processing of organic carbon, and therefore these longer term fates are not directly relevant to the categorisation. The wording of section 4.5 has been adapted to clarify this.

The reviewer is also correct that smaller variations in the relative importance of different pathways 31 tend to be observed within the short term, however this does not lead to problems for our

- 32 categorisation. The experiments presented in figure 5 range from 6 h to 23 days, with the majority
- falling in the 1-7 days range (i.e. the single 23 day experiment was the only one longer than 7 days).
- 34 Therefore the only one which cannot truly be said to represent 'short-term' biological C processing is the 23 day experiment (Porcupine Abyssal Plain, Witte et al., 2003b). This has been excluded.

In a few cases experiments were conducted over multiple durations at the same sites. In the case of sites across the Pakistan margin the difference in duration between 2 and 5 days never caused a shift in the category of short term biological C processing (Woulds et al., 2009). Similarly in the

Sognefjord the C processing pattern remained in the same category in experiments lasting both 1.5

and 3 days (Witte et al., 2003a). In the German Bight, experiments lasting 0.5 to 1.5 days always showed a bacterial uptake dominated pattern, and bacterial uptake remained equally important as respiration after 5.5 days (Bhuring et al., 2006). Therefore, while we accept that experiment duration does play a role in determining the finer detail of the pattern of biological C processing observed in
 an experiment, it does not determine the category of C processing pattern within the range of experiment durations included here (and is certainly not the 'most important factor' as the reviewer suggests).

The Porcupine Abyssal Plain is the only example of a site where different short-term experiment 48 durations led to different biological C processing categories (Witte et al., 2003b). At this site, where 49 we would expect to see 'respiration dominated' biological C processing, the shortest experiment (60 50 h) actually showed 'active faunal uptake', with macrofaual uptake accounting for 26% of biological C 51 processing. All longer experiments (8 d and 23 d) showed 'respiration dominated' biological C 52 processing. This site has been removed from the standard categorisation and is instead discussed 53 alongside the other exceptions.

Therefore we feel that the variation in experiment duration between the results does not cause 55 sufficient changes to C processing patterns to invalidate the categorisation, and that therefore 56 section 4.5 and figure 5 should be retained. We have added discussion of effect of experiment 57 duration on categorisation as part of the discussion of the Porcupine Abyssal Plain experiments 58 (detailed above), and have added a column to table 1 showing experiment duration, so that all 59 details are clearly available.

Reviewer: Differences in light condition. The authors performed the 13C-labeled phytodetritus 61 experiments with and without light (with light: Loch Etive, without light: Ythan sand flat). The 62 authors validate the different conditions because natural environments are dark and light 63 conditions, respectively. However, I believe that the incubation with light makes complicated pathways. Without light, the 13C-phytodetritus is ncorporated into heterotrophic microbes or 64 65 eukaryotes, and either assimilated into their biomass or respired as 13CO2. With light, however, the 66 respired 13CO2 can be assimilated into photoautotrophic microbial biomass via photosynthesis. This 67 leads underestimation of respired carbon and overestimation of bacterial assimilation. Without light, 68 chemolithoautotrophic microbes can also cause same process, but the contribution must be smaller 69 than photosynthesis. How much proportion of CO2 was labeled with 13C? If the 13C concentrations 70 in CO2 is almost negligible (few %), then the bacterial assimilation via photosynthesis may also be 71 negligible. This can be calculated from the DIC-d13C data of the study. Or, if there are literature which investigated bacterial community at this area, then the authors may validate that 72 73 photoautotrophic bacteria was minor.

Answer: Once again the reviewer is correct that the different light conditions led to a difference in 75 the C flow pathways that were possible in the two experiments. However the different light levels 76 were necessary in order to correctly re-create natural conditions. The labelling level of DIC in the 77 Ythan experiment remained very low throughout (never 1.33> atom % 13C), therefore the 78 underestimation of respiration due to use of respired DIC by photoautotrophs is negligible, as the 79 reviewer suggests. In addition, this will not have interfered with measurements of bacterial C uptake 80 as the sub-set of PLFAs used are specific to bacteria (as opposed to benthic algae), and are regularly used for this purpose, including in intertidal incubations performed in the presence of light. A note 81 82 has been added to section 4.1. 83

*Reviewer:* Uptake calculation The authors calculated the Carbon uptake by sample with the equation
 (3), line 253. However, the At% phytodetritus must be subtracted by At% background. I understand
 that the extent of 13C-label in this study (25% and 34%) are high and the re-calculated values using
 subtracted value may change only 2 or 3 % (considering 25 become 23.9 and 34 become 32.9).
 However, the it is necessary to indicate appropriate values as much as possible.

Answer: We do not agree that it is necessary to subtract the natural occurrence of 13C from the
 labelling level of the phytodetritus when calculating C uptake into the different C pools. It is true that
 phytodetritus grown without any artificial 13C enrichment would indeed have contained a natural
 amount of 13C, but this does not change the fact that the phytodetritus actually added to our
 experiments had the labelling levels as measured and reported. Both the 'naturally' present and
 artificially enriched fractions of the 13C in the phytodetritus serve as tracer, and the only thing that
 has to be subtracted out is naturally occurring 13C in the sediment system to which the tracer was added. We do not feel that it is necessary to add this explanation to the manuscript, unless theeditor feels that it should go in.

- 97 Specific comments:
- 98 Reviewer: Line 32 Did the accessibility by bacteria to added C similar between two sites? Please
show the vertical profiles of 13C if possible.

**Answer:** Accessibility by bacteria will have been similar in the sense that in both experiments

- phytodetritus was added to the sediment surface in the same way. Thereafter it may have been
   transported through the sediment differently due to differences between permeable and cohesive
- 103 sediments. Unfortunately downcore 13C profiles are not available.

Reviewer:Line 145. Figure 1 does not show any sills or geographical names. Please include these
 information to the figure or delete the citation (Fig.1) from the end of this sentence.

Answer: Figure reference removed.

- 109 **Reviewer:**Line 163. While the Loch Evive site has 70 m water depth, the Ythan estuary site exposed
- during low tide. This is a great difference between two sites, in addition to sediment grain size and OC concentrations. The authors need to discuss the potential impacts of these differences of OC
- cycling and validate why the authors did not perform the experiment at coarse grained, OC poor site
- 113 having similar water depths (or vice versa).
- Answer: This was driven by the coring technique and technology available for coarse grained
   sediment (required taking cores by hand). A note has been added to section 4.1.
- 116
- **Reviewer:**Line 171. What exactly was the phytodetritus labeled with 13C? Was that degraded in some way? Or some sort of algal species? Was this same to the one which was added to Ythan sand
- 119 flat? Please clarify these details.
- 120 **Answer:** Details have been added in the methods section.
- 121
- 122 **Reviewer:**Line 173. How much volume was the overlying water in the core?
- 123 Answer: Detail added.

- Reviewer:Line 185 150 um sieve is not typical size separation for meiofauna. Why did the authorschoose this size?
- Answer: It was only practical to extract the larger meiofauna, as time did not permit sorting fauna all
   the way down to 63μm. Such small fauna would also have been very challenging to analyse. A note
   has been added.

Reviewer:Line 189 Why the authors used milliQ water instead filtered seawater of artificial 131 132 seawater? MilliQ water may had elution of organic matters from fauna due to osmoticshock (although the results showed insignificant effect). 133 134 Answer: Filtered seawater was used, not milliQ. This has been corrected. 135 136 Reviewer:Line 196 Bubbling with air in this experiment while the Loch Etive site cores 137 weremaintained with oxystat system. How did this affect to 13C-CO2 amounts? 138 Answer: There will not have been an effect from air bubbling in the Ythan experiment on measured respiration rates, as air bubbling did not occur during respiration measurement periods, and  $^{\rm 13}{\rm C}\,{\rm DIC}$ 139 data from outside of those periods was not used in respiration calculations. Potential effects of the 140 141 oxystat system on respiration measurements in the Loch Etive experiment are already addressed in 142 the manuscript as follows "As the tubing used in the oxystat gill was permeable to all gases there was the potential for loss of some 13CO2 generated during the experiment. However, the dissolved 143 inorganic carbon (DIC) concentration difference between the incubation water and oxygenated 144 145 reservoir will have remained small, thus this effect is thought to be minor. " 146 147 Reviewer:Line 253 The equation is not presented in correct way (no under bar below "C 148 Uptakesample". What the unit of "C Uptake sample"? 149 Answer: Equation has been corrected, and units added. 150 Reviewer: Line 263 It is not clear about the linear regression. Do the authors mean linear 151 152 regression f different incubation periods? It is also important to show the changes in d13C-DIC (or 153 13C-respiration rates) with time, because the changes in 13C-respiration with time should give 154 crucial info regarding faunal or bacterial responses and C processing. 155 Answer: Respiration was calculated separately for each separate incubation period. 156 157 Reviewer:Line 267 It is necessary to show the respiration data of Ythan sand flat, too, as Tableor 158 supplementary figure. 159 Answer: A supplementary figure has been added displaying the increase in labelled DIC over time for 160 all chambers, and including regression lines and equations. This has been referred to in the text as 161 appropriate. 162 Reviewer:Line 274 Please describe the centrifuge condition ( x g, how long, and what 163 164 temperatureetc). It will help to guess the potential effects of centrifuge on bacterial PLFA loss. Answer: Detail has been added. 165 166 167 Reviewer:Line 279. Did the authors examine the d13C of bulk sediments? If so, please includeas 168 Table etc.

- 169 Answer: These data are not available.
- 170 Reviewer: Line 282 Again, it is important to show temporal changes in d13C (or respired 13C).
- 171 **Answer:** A supplementary figure has been added.
- Reviewer: Line 326. 0.00023 mgC per mgC corresponds \_5 or 10 per mil of Dd13C, which isrelatively
   low labeling. What were the variation in d13C of natural PLFA and labeled
- 174 PLFA? Can you add as Table?

Answer: This is a large amount of data to tabulate (del13C values for several depths in the sediment,
plus background values, for 4 PLFAs, for each of 4 incubation cores per site), and I am not convinced
that it would provide much clarity for the reader. The background del13C values for the bacteriaspecific PLFAs were similar at each site (-20 to -25 ‰). Δδ values were higher than the reviewer
suggests in the surface sediment horizon (100's ‰), but this will have been balanced by them being
lower (10's ‰ or less) in deeper horizons at Loch Etive. As expected, these Δδ values were at least an order of magnitude greater for the Ythan sand flat.

Reviewer:Lines 347 to 353. Whatever the C dose amounts were similar, the authors should think
 about the difference in natural phytodetritus supply rates at two sites. The same amount of 13C phytodetritus input should have completely different effects on between originally eutrophic (in
 terms of OM) site and oligotrophic site. The authors should discuss these point of view by referring the primary production rates at two sites.

Answer: We acknowledge that the C dose represented a different proportion of naturally present OC at each site, and this could have led to an enhanced response at the Ythan sand flat. However, surface sediment OC concentrations are not necessarily a good reflection of actual C delivery to the seafloor, given the different transport mechanisms in permeable and cohesive sediments (see discussion). Further, there is a sparsity of data available on primary production rates, particularly for the Ythan sand flat. Therefore maintaining a uniform C addition was judged to yield the most comparable data. This discussion has been added.

- 196 **Reviewer:**Line 368 Can you cite any paper which dealing different size screens?**Answer:** We are not
- 197 aware of a paper dealing with the effect caused by this difference in screen sizes, and can only re-
- 198 iterate that the sizes were standard for the sediment types in question, and were also most
- 199 favourable in terms of practicality (using a 300µm screen in a sandy sediment would lead to very
- 200 high retention of sediment, making extraction of fauna particularly difficult).

- Reviewer: Lines 376 to 380. Due to the osmotic shock by milliQ water (according to M&M), the
   fauna may be dead and did not have a time to void the gut.
- Answer: This step was not actually conducted in Milli-Q (corrected in response to an earlier
   comment), so osmotic shock will not have been a problem.

Reviewer: Line 431. Gooday et al. 2008 represent biomass-uptake relationships with different
 symbols for bacteria, fauna, foraminifera. Can you also make such kind of Figure 4 for better
 comparison?

Answer: Figure 4 would be unclear if taxonomic information was included where all the points are
 plotted together, therefore two panels have been added, one for each site, showing data for the
 different taxa.

- Reviewer: Line 438. This may suggest that the macrofauna of Ythan sand flat has low background
   metabolism than Loch Etive.
- 216 Answer: Agreed, this comment has been added.

Reviewer: Line 459. I cannot follow why the authors said "macrofaunal biomass" in this sentence
 whereas the line 456 mentioned "biomass (faunal plus bacterial)". Please describe

- 220 more in detail if the authors actually intended to say "macrofaunal biomass".
- 221 Answer: Clarification has been added.

- 223 **Reviewer:** Chapter 4.4. can be combined to 4.3.
- 224 Answer: These two sections both consider points related to faunal C uptake. However, the main
- point made in section 4.4. is distinct from those in section 4.3, and therefore we feel that the
- additional sub-heading remains helpful.

Reviewer: Line 520. Both methods (Total respiration rate measurements and bacterial C assimilation rates) has considerable uncertainty. Thus the discussion here, dealing bacterial growth efficiency, is somewhat over-interpretation. Also, as mentioned earlier, because the incubation of Ythan sand flat sediment was carried out under light condition, it is possible that some 13C-bacterial lipids were originated from the photoautotrophic microbes, not by heterotrophic bacteria which incorporated 13C-labeled phytoplankton.

Answer: The sub-set of PLFAs used to quantify bacterial uptake are regularly used to indicate bacterial activity as separate from microphytobenthis production, including in incubations in which
 light was present. We agree however, and acknowledge in the text, that our measurements do not
 allow an accurate quantification of bacterial growth efficiency. The text has been shortened accordingly.

- Reviewer: Line 571 Again, temporal changes in DIC-13C at both site may give better idea aboutthese interpretations.
- 242 **Answer:** A supplementary figure has been added.

**Reviewer:** Line 673 Hunter et al. 2012b. There is no Hunter et al. 2012a, thus deleted "b".

- Answer: Corrected.
- 246
- 247 **Reviewer:** Table 1 Please add a new column showing incubation periods.
- 248 Answer: Added.
- 249 Reviewer: Figure 2. Please add "n.d." for meiofauna and foraminifera of Ythan sand flat.
- 250 Answer: Note added to the caption.
- 251

**252 Reviewer 2:**

- 253 Once again we would like to thank the reviewer for their overall positive opinion, and for their
- attention to detail which will allow us to improve the manuscript.

**255 Major comments:**

- 256 The main comment from this reviewer is that the discussion is overly long. We agree, especially
- concerning the section about bacterial growth efficiency. The discussion has now been shortenedsignificantly.

**259 Specific comments:**

- Reveiwer: Line 73: It might be worth pointing out what does biological C processing not cover. Is
   there non-biological C processing in these systems? It might be worth pointing out the differences.
- Answer: The term is used to distinguish between short term uptake and cycling and longer tern C
   burial. This has been clarified.

Reveiwer: Line 76: A quibble: Stable isotope tracer experiments are an excellent tool, but not ideal.
 For instance, radiotracer 14C incubations are far more sensitive and do not depend on sorting out
 mass of naturally occurring background tracer distribution.

- Answer: Acknowledged, but working with stable isotopes has practical benefits which can allow
   increased numbers of experimental treatments/durations/replicates. Wording has been changed.
- Reveiwer: Line 117 and following: Independent of the food-web tracer studies, it would be nice to
   have some information on the relative benthic biomasses for these two sediment types, e.g. muddy
   and sandy bottoms. I would be surprised if muddy bottoms actually supported more faunal biomass.
- 272 **Answer:** This section does not seem an appropriate place to review biomass data for different
- estuarine sediments, however such details can be found in Table 1 (biomass data are independent of
   the associated C tracing experiments)..
- Reveiwer: With the exception of the respiration measurements, these are single endpoint
   experiments. Dynamics between the pools are not necessarily accessible.
- 277 Answer: We are not clear which part of the text the reviewer is referring to here.
- 278 Reveiwer: Line 124: "Recent findings" is relative; dynamic biogeochemical cycling in low OC
- 279 permeable sediments has been extensively documented over the last two decades.
- 280 Answer: Agreed, wording has been amended.

- *Reveiwer:* Line 171: Please describe more carefully the labeled phytodetritus in more detail. Was it
   composed of a single species and what? Was it prepared in the same fashion for both sites? What
- 283 was it composed of? How fresh was it? Was it added as fresh or freeze-dried material.
- 284 Answer: Detail has neen added to the methods section.
- **Reveiwer:** Does the difference between the labeling percentages (ca. 25% and 34%) for the two sites reflect different batch preparations, or differing compositions of pytodetritus?
- 287 Answer: Different phytodetritus batches and species. Detail has been added (see above)..
- *Reveiwer:* Methods: It's not entirely clear to me that total bulk 13C of the sediment was determined
   (i.e. total Corg 13C). This must have been done in order to calculate the recoveries of tracers shown
   in Figure 2.
- Answer: The totals shown in figure 2 are total biologically processed C, and therefore do not contain
   C remaining in the sediment. Data for 13C remaining in the sediment are not available.
- 293 **Reveiwer:** Is there a time zero sample, i.e. samples taken from one core immediately after the 294 addition of the 13C-labeled phytodetritus?
- Answer: This is only available for Loch Etive, and not on the Ythan sand flat, therefore data has not been included.
- 297 **Reveiwer:** Line 244 and following: It is not really clear to me why the authors work with the del (\_)
- notation for these type of experiments. There is also no obvious connection from how they go from
   Equation 2 to Equation 3, the latter of which is the more relevant for this manuscript.
- **Answer:** Data are reported using the del notation in the results section because many workers in the field use this notation, and  $\Delta\delta$  is a clear way of displaying isotopic enrichments. However, our
- 302 calculations for uptake used At% instead. There is not supposed to be a connection between
- 303 equation 2 and equation 3.
- Reveiwer: Calculations with exceedingly large enrichments, for instance as seen in the macrofaunal
   biomass (lines 290 and following), become inaccurate.
- Answer: The reviewer's meaning is not quite clear, however if this is given as a reason for not using
   the del notation, then note that uptake calculations were made using At% instead.
- 308 *Reveiwer:* Line 280: . . . . or as dissolved organic carbon.
- 309 Answer: This has been added.
- Reveiwer: Section 3.1: It might be helpful for the reader to plot the remineralization data over the
   time course of the experiment.
- 312 Answer: These plots have been added as supplementary information.
- 313 Reveiwer: Section 4.2: This whole discussion is rather contradictory. On one hand the authors claim
- 314 that the temperature and organic C loading are similar (line 384), but then suggest that temperature
- 315 plays a larger role than biomass or organic C (lare 395) does not make sense. Furthermore, there are
- 316 no proper controls for assessing any of these factors. I would drop this whole discussion (see further 317 discussion about curtailing discussion) and the conclusions regarding temperature (line 571). This
- 318 was not the point of the study, it was not properly assessed, nor is it supported by the data.
- 319 **Answer:** The point made in this section is that despite many differences between the settings of the
- 320 two experiments, the measured respiration rates were very similar, and this is attributed to the fact
- 321 that the experiments were conducted at the same temperature. Supporting material from the
- 322 literature is provided. This section has been shortened and clarified, and we do not feel that this
- 323 point goes beyond the reach of the data.

**Reveiwer:** Line 494: "This hypothesis. . .." Which hypothesis? From this paper or Woulds et al. 2009?

Actually I find the whole discussion of hypotheses, both here and earlier in the manuscript (line 134
 and following) a bit specious. I think that it is enough for the authors to state that they are comparing two types of sites that are thus far lacking from the overall range of sites on which such
 experiments have been performed.

Answer: The sentence in question has been changed to clarify which hypothesis is being referred to.It is not clear why the reviewer finds the earlier statement of our hypotheses to be specious, as the reflect our expectations before the experiments were conducted. We have retained these hypotheses, as we feel they are the best way of providing the appropriate focus to the manuscript.

Reveiwer: Figure 2: I assume that "Total" stand for the sum of respiration, bacterial uptake, etc. in
 that case, there is also no Total for the % of Biologically Processed C. It might also be interesting to
 add a third panel to include total initial pool size of each of the separate pools (i.e. how much C is in
 each pool originally).

Answer: This is an interesting suggestion, but we do not feel that it would work well in graph form. The information on macrofaunal and bacterial biomass is already given in the results text. It is not so helpful to quantify the amount of C in the DIC pool, as the absolute amount depends on the height of the water column in the experimental chamber. This varied for each chamber. Of course it could be given for a standard height of water column, but the choice of height would be entirely arbitrary, and so it would not make a meaningful comparison with the macrofaunal and bacterial pools.

Reveiwer: Figures 5 and 6: These figures are quite compelling, although I think that they could be
 combined. It is not entirely obvious why there are two separate figures.

**Answer:** The figures have been combined.

[revised manuscript text omitted]
|     | D                                                                                                                      |
| 594 | $\delta \mathscr{H}_o = \left(\frac{Ks}{Rr} - 1\right) x \ 1000$                                                       |

Where Rs and Rr are the 13C/12C ratio in the sample and the reference standard respectively. Isotopic enrichments ( $\Delta\delta$ ) were then calculated using Eq. (2).

(1)

(2)
Carbon uptake by faunal groups was calculated by subtracting naturally occurring 13C, multiplying by the
sample C contents, and correcting for the fact that the added phytodetritus was not 100 % 13C labelled, as shown
in Eq. (3):

 $\Delta \delta = \delta^{13} C \ sample - \ \delta^{13} C background$

 $\frac{C \ Uptake_{sample} = (At \ \%_{sample} - At \ \%_{background}) X \ C \ Contents_{sample}}{At \ \%_{phytodetritus}} X100$

 $C \ Uptake_{sample} (\mu g) = \frac{\left(At \ \%_{sample} - At \ \%_{background}\right) \times C \ Contents_{sample}}{At \ \%_{phytodetritus}} \times 100$

Where At % is the 13C atoms present as a percentage of the total C atoms present. Data from individual 607 specimens was summed to produce faunal C uptake by different groups of fauna. For Loch Etive, background 608 13C was subtracted based on natural faunal isotopic data collected concurrently with the C tracing experiment. 609 For the Ythan sand flat natural faunal isotopic data were not available, and instead the natural C isotopic 610 signature of sedimentary organic C (-20.2 ‰) was used. Isotopic signatures of fauna at the end of the 611 experiment had a maximum of 2460‰ and a mean of 175‰. Therefore the small inaccuracies introduced by the 612 use of this natural background value will not have been significant.

The DIC concentrations and  $\delta^{13}$ C-DIC were used to calculate the total amount of added  $^{13}$ C present as DIC in experimental chambers at each sampling time. A linear regression was applied to these to yield a separate respiration rate for each core and for each period of respiration measurement (mean  $R^2 = 0.909$ , with the exception of one measurement showing poor linearity with  $R^2 = 0.368$ ), and the rate was multiplied by experiment duration to calculate total respiration of added C during the experiment. In the case of the Ythan sand flat respiration was measured during two separate 24 h periods through the experiment. In this case averagerates from the two measurements were used to calculate total respiration of added C throughout the experiment.

Bacterial C uptake was quantified using the compounds iC14:0, iC15:0, aiC15:0 and iC16:0 as bacterial markers. Bacterial uptake of added C was calculated from their concentrations and isotopic compositions (corrected for natural 13C occurrence using data from unlabelled sediment), based on these compounds representing 14% of total bacterial PLFAs, and bacterial PLFA comprising 5.6% of total bacterial biomass (Boschker and Middelburg, 2002). In the case of Loch Etive, the sediments from which PLFAs were extracted had previously been centrifuged (10 mins, 3500 rpm, room temperature) for porewater extraction, which could have led to a slight reduction in the bacterial biomass and C uptake measured.

**627 3. Results**

The mean recovery of added C from the bacterial, faunal and respired pools together was 30±6% and 31±10%
of that which was added for Loch Etive and the Ythan sand flat respectively. This is a good recovery rate compared to other similar experiments (e.g. Woulds et al., 2007). Most of the remaining C was likely left in the
sediment as particulate organic C or as dissolved organic C.

**632 3.1 Remineralisation**

The average respiration rate of the added OC was similar in Loch Etive and the Ythan sand flat, and reached 634 0.64±0.4 and 0.63±0.12 mg C m-2h-1, respectively. Thus, the total amount of added C that was respired at each 635 site (over 156 h in Loch Etive and 162 h on the Ythan sand flat) was 99.5±6.5 and 102.6±19.4 mg C m-2 for 636 Loch Etive and the Ythan sand flat, respectively (Fig. 2). In both experiments, respiration rates measured in the 637 first 48 h (1.41±0.14 and 0.74±0.02 mg C m-2h-1 for Etive and the Ythan sand flat, respectively) were higher 638 than those measured in the last 48 h of the experiment (0.31±0.04 and 0.52±0.22 mg C m-2h-1 for Etive and the (3)

**Y than sand flat, respectively; this difference was significant only for Loch Etive, t-test, P<0.001). The increase in labelled DIC over time for each chamber is shown in Fig. S1.**

**641 3.2 Faunal biomass and C uptake**

 $642 \qquad \text{Macrofaunal biomass in the experimental cores was } 4337 \pm 1202 \text{ mg C m}^{-2} \text{ in Loch Etive and } 455 \pm 167 \text{ 
[revised manuscript text omitted]

| 1012 | bernario, s. 1, E. an r. Dradoe, r., hoodberr, E., isleer isles, r. a wirre, c. 2000. Benare                                                                                                                                                                                                                                                                                                                                                                                                                                                                                                                                                                                                                                                                                                                                                                                                                                                                                                                                                                                                                                                                                                                                                                                                                                                                                                                                                                                                                                                                                                                                                                                                                                                                                                                                                                                                                                                                                                                                                                                                                                   |  |  |  |  |  |  |  |
|------|--------------------------------------------------------------------------------------------------------------------------------------------------------------------------------------------------------------------------------------------------------------------------------------------------------------------------------------------------------------------------------------------------------------------------------------------------------------------------------------------------------------------------------------------------------------------------------------------------------------------------------------------------------------------------------------------------------------------------------------------------------------------------------------------------------------------------------------------------------------------------------------------------------------------------------------------------------------------------------------------------------------------------------------------------------------------------------------------------------------------------------------------------------------------------------------------------------------------------------------------------------------------------------------------------------------------------------------------------------------------------------------------------------------------------------------------------------------------------------------------------------------------------------------------------------------------------------------------------------------------------------------------------------------------------------------------------------------------------------------------------------------------------------------------------------------------------------------------------------------------------------------------------------------------------------------------------------------------------------------------------------------------------------------------------------------------------------------------------------------------------------|--|--|--|--|--|--|--|
| 1013 | microbial and whole-community responses to different amounts of 13 C-enriched algae: In situ experiments in the                                                                                                                                                                                                                                                                                                                                                                                                                                                                                                                                                                                                                                                                                                                                                                                                                                                                                                                                                                                                                                                                                                                                                                                                                                                                                                                                                                                                                                                                                                                                                                                                                                                                                                                                                                                                                                                                                                                                                                                                     |  |  |  |  |  |  |  |
| 1014 | deen Croten Sea (Eastern Mediterranean) Limnology and Oceanography 51, 157, 165                                                                                                                                                                                                                                                                                                                                                                                                                                                                                                                                                                                                                                                                                                                                                                                                                                                                                                                                                                                                                                                                                                                                                                                                                                                                                                                                                                                                                                                                                                                                                                                                                                                                                                                                                                                                                                                                                                                                                                                                                                                |  |  |  |  |  |  |  |
| 1011 | deep eretain bea (Eastern Mediternalean). Eminology and oceanography, 51, 157-165.                                                                                                                                                                                                                                                                                                                                                                                                                                                                                                                                                                                                                                                                                                                                                                                                                                                                                                                                                                                                                                                                                                                                                                                                                                                                                                                                                                                                                                                                                                                                                                                                                                                                                                                                                                                                                                                                                                                                                                                                                                             |  |  |  |  |  |  |  |
| 1015 | CANFIELD, D. E. 1994. Factors influencing organic carbon preservation in marine sediments. Chemical                                                                                                                                                                                                                                                                                                                                                                                                                                                                                                                                                                                                                                                                                                                                                                                                                                                                                                                                                                                                                                                                                                                                                                                                                                                                                                                                                                                                                                                                                                                                                                                                                                                                                                                                                                                                                                                                                                                                                                                                                            |  |  |  |  |  |  |  |
| 1016 | Geology 114 315-329                                                                                                                                                                                                                                                                                                                                                                                                                                                                                                                                                                                                                                                                                                                                                                                                                                                                                                                                                                                                                                                                                                                                                                                                                                                                                                                                                                                                                                                                                                                                                                                                                                                                                                                                                                                                                                                                                                                                                                                                                                                                                                            |  |  |  |  |  |  |  |
| 1010 | 000053, 111, 010-022.                                                                                                                                                                                                                                                                                                                                                                                                                                                                                                                                                                                                                                                                                                                                                                                                                                                                                                                                                                                                                                                                                                                                                                                                                                                                                                                                                                                                                                                                                                                                                                                                                                                                                                                                                                                                                                                                                                                                                                                                                                                                                                          |  |  |  |  |  |  |  |
| 1017 | DUARTE, C. M., MIDDELBURG, J. J. & CARACO, N. 2005, Major role of marine vegetation on the oceanic                                                                                                                                                                                                                                                                                                                                                                                                                                                                                                                                                                                                                                                                                                                                                                                                                                                                                                                                                                                                                                                                                                                                                                                                                                                                                                                                                                                                                                                                                                                                                                                                                                                                                                                                                                                                                                                                                                                                                                                                                             |  |  |  |  |  |  |  |
| 1010 |                                                                                                                                                                                                                                                                                                                                                                                                                                                                                                                                                                                                                                                                                                                                                                                                                                                                                                                                                                                                                                                                                                                                                                                                                                                                                                                                                                                                                                                                                                                                                                                                                                                                                                                                                                                                                                                                                                                                                                                                                                                                                                                                |  |  |  |  |  |  |  |
| 1018 | carbon cycle.Biogeosciences, 2, 1-8.                                                                                                                                                                                                                                                                                                                                                                                                                                                                                                                                                                                                                                                                                                                                                                                                                                                                                                                                                                                                                                                                                                                                                                                                                                                                                                                                                                                                                                                                                                                                                                                                                                                                                                                                                                                                                                                                                                                                                                                                                                                                                           |  |  |  |  |  |  |  |
| 1019 | EHRENHAUSS S & HUETTEL M 2004 Advective transport and decomposition of chain-forming planktonic                                                                                                                                                                                                                                                                                                                                                                                                                                                                                                                                                                                                                                                                                                                                                                                                                                                                                                                                                                                                                                                                                                                                                                                                                                                                                                                                                                                                                                                                                                                                                                                                                                                                                                                                                                                                                                                                                                                                                                                                                                |  |  |  |  |  |  |  |
| 1020 |                                                                                                                                                                                                                                                                                                                                                                                                                                                                                                                                                                                                                                                                                                                                                                                                                                                                                                                                                                                                                                                                                                                                                                                                                                                                                                                                                                                                                                                                                                                                                                                                                                                                                                                                                                                                                                                                                                                                                                                                                                                                                                                                |  |  |  |  |  |  |  |
| 1020 | diatoms in permeable sediments. Journal of Sea Research, 52, 1/9-19/.                                                                                                                                                                                                                                                                                                                                                                                                                                                                                                                                                                                                                                                                                                                                                                                                                                                                                                                                                                                                                                                                                                                                                                                                                                                                                                                                                                                                                                                                                                                                                                                                                                                                                                                                                                                                                                                                                                                                                                                                                                                          |  |  |  |  |  |  |  |
| 1021 | EVRARD V SOFTAERT K HEIR C H R HUETTEL M YENOPOULOS M A & MIDDELRURG L                                                                                                                                                                                                                                                                                                                                                                                                                                                                                                                                                                                                                                                                                                                                                                                                                                                                                                                                                                                                                                                                                                                                                                                                                                                                                                                                                                                                                                                                                                                                                                                                                                                                                                                                                                                                                                                                                                                                                                                                                                                         |  |  |  |  |  |  |  |
| 1021 |                                                                                                                                                                                                                                                                                                                                                                                                                                                                                                                                                                                                                                                                                                                                                                                                                                                                                                                                                                                                                                                                                                                                                                                                                                                                                                                                                                                                                                                                                                                                                                                                                                                                                                                                                                                                                                                                                                                                                                                                                                                                                                                                |  |  |  |  |  |  |  |
| 1022 | J. 2010. Carbon and nitrogen flows through the benthic food web of a photic subtidal sandy sediment. Marine                                                                                                                                                                                                                                                                                                                                                                                                                                                                                                                                                                                                                                                                                                                                                                                                                                                                                                                                                                                                                                                                                                                                                                                                                                                                                                                                                                                                                                                                                                                                                                                                                                                                                                                                                                                                                                                                                                                                                                                                                    |  |  |  |  |  |  |  |
| 1023 | Ecology-Progress Series, 416, 1-16.                                                                                                                                                                                                                                                                                                                                                                                                                                                                                                                                                                                                                                                                                                                                                                                                                                                                                                                                                                                                                                                                                                                                                                                                                                                                                                                                                                                                                                                                                                                                                                                                                                                                                                                                                                                                                                                                                                                                                                                                                                                                                            |  |  |  |  |  |  |  |
|      |                                                                                                                                                                                                                                                                                                                                                                                                                                                                                                                                                                                                                                                                                                                                                                                                                                                                                                                                                                                                                                                                                                                                                                                                                                                                                                                                                                                                                                                                                                                                                                                                                                                                                                                                                                                                                                                                                                                                                                                                                                                                                                                                |  |  |  |  |  |  |  |
| 1024 |                                                                                                                                                                                                                                                                                                                                                                                                                                                                                                                                                                                                                                                                                                                                                                                                                                                                                                                                                                                                                                                                                                                                                                                                                                                                                                                                                                                                                                                                                                                                                                                                                                                                                                                                                                                                                                                                                                                                                                                                                                                                                                                                |  |  |  |  |  |  |  |
| 1025 |                                                                                                                                                                                                                                                                                                                                                                                                                                                                                                                                                                                                                                                                                                                                                                                                                                                                                                                                                                                                                                                                                                                                                                                                                                                                                                                                                                                                                                                                                                                                                                                                                                                                                                                                                                                                                                                                                                                                                                                                                                                                                                                                |  |  |  |  |  |  |  |
| 1025 | EVKAKD, V., HUETTEL, M., COUK, P. L. M., SOETAERT, K., HEIP, C. H. K. & MIDDELBURG, J. J.                                                                                                                                                                                                                                                                                                                                                                                                                                                                                                                                                                                                                                                                                                                                                                                                                                                                                                                                                                                                                                                                                                                                                                                                                                                                                                                                                                                                                                                                                                                                                                                                                                                                                                                                                                                                                                                                                                                                                                                                                                      |  |  |  |  |  |  |  |
| 1026 | 2012. Importance of phytodetritus and microphytobenthos for heterotrophs in a shallow subtidal sandy                                                                                                                                                                                                                                                                                                                                                                                                                                                                                                                                                                                                                                                                                                                                                                                                                                                                                                                                                                                                                                                                                                                                                                                                                                                                                                                                                                                                                                                                                                                                                                                                                                                                                                                                                                                                                                                                                                                                                                                                                           |  |  |  |  |  |  |  |
| 1027 | sediment. Marine Ecology Progress Series, 455, 13-31.                                                                                                                                                                                                                                                                                                                                                                                                                                                                                                                                                                                                                                                                                                                                                                                                                                                                                                                                                                                                                                                                                                                                                                                                                                                                                                                                                                                                                                                                                                                                                                                                                                                                                                                                                                                                                                                                                                                                                                                                                                                                          |  |  |  |  |  |  |  |
|      |                                                                                                                                                                                                                                                                                                                                                                                                                                                                                                                                                                                                                                                                                                                                                                                                                                                                                                                                                                                                                                                                                                                                                                                                                                                                                                                                                                                                                                                                                                                                                                                                                                                                                                                                                                                                                                                                                                                                                                                                                                                                                                                                |  |  |  |  |  |  |  |
| 1028 | GAGE, J. D. 1972. Community structure of the benthos in Scottish sea-lochs. I Introduction and species                                                                                                                                                                                                                                                                                                                                                                                                                                                                                                                                                                                                                                                                                                                                                                                                                                                                                                                                                                                                                                                                                                                                                                                                                                                                                                                                                                                                                                                                                                                                                                                                                                                                                                                                                                                                                                                                                                                                                                                                                         |  |  |  |  |  |  |  |
| 1029 | diversity.Journal of Marine Biology, 14, 281-297.                                                                                                                                                                                                                                                                                                                                                                                                                                                                                                                                                                                                                                                                                                                                                                                                                                                                                                                                                                                                                                                                                                                                                                                                                                                                                                                                                                                                                                                                                                                                                                                                                                                                                                                                                                                                                                                                                                                                                                                                                                                                              |  |  |  |  |  |  |  |
|      |                                                                                                                                                                                                                                                                                                                                                                                                                                                                                                                                                                                                                                                                                                                                                                                                                                                                                                                                                                                                                                                                                                                                                                                                                                                                                                                                                                                                                                                                                                                                                                                                                                                                                                                                                                                                                                                                                                                                                                                                                                                                                                                                |  |  |  |  |  |  |  |
| 1030 | DEL GIORGIO, P. A. & COLE, J. J. 1998.Bacterial growth efficiency in natural aquatic systems. Annual                                                                                                                                                                                                                                                                                                                                                                                                                                                                                                                                                                                                                                                                                                                                                                                                                                                                                                                                                                                                                                                                                                                                                                                                                                                                                                                                                                                                                                                                                                                                                                                                                                                                                                                                                                                                                                                                                                                                                                                                                           |  |  |  |  |  |  |  |
| 1031 | Review of Ecology and Systematics, 29, 503-541.                                                                                                                                                                                                                                                                                                                                                                                                                                                                                                                                                                                                                                                                                                                                                                                                                                                                                                                                                                                                                                                                                                                                                                                                                                                                                                                                                                                                                                                                                                                                                                                                                                                                                                                                                                                                                                                                                                                                                                                                                                                                                |  |  |  |  |  |  |  |
|      |                                                                                                                                                                                                                                                                                                                                                                                                                                                                                                                                                                                                                                                                                                                                                                                                                                                                                                                                                                                                                                                                                                                                                                                                                                                                                                                                                                                                                                                                                                                                                                                                                                                                                                                                                                                                                                                                                                                                                                                                                                                                                                                                |  |  |  |  |  |  |  |
| 1032 | GONTIKAKI, E., MAYOR, D. J., NARAYANASWAMY, B. E. & WITTE, U. 2011. Feeding strategies of                                                                                                                                                                                                                                                                                                                                                                                                                                                                                                                                                                                                                                                                                                                                                                                                                                                                                                                                                                                                                                                                                                                                                                                                                                                                                                                                                                                                                                                                                                                                                                                                                                                                                                                                                                                                                                                                                                                                                                                                                                      |  |  |  |  |  |  |  |
| 1033 | deep-sea sub-Arctic macrofauna of the Faroe-Shetland Channel: Combining natural stable isotopes and                                                                                                                                                                                                                                                                                                                                                                                                                                                                                                                                                                                                                                                                                                                                                                                                                                                                                                                                                                                                                                                                                                                                                                                                                                                                                                                                                                                                                                                                                                                                                                                                                                                                                                                                                                                                                                                                                                                                                                                                                            |  |  |  |  |  |  |  |
| 1034 | enrichment techniques. Deep-Sea Research Part I-Oceanographic Research Papers, 58, 160-172.                                                                                                                                                                                                                                                                                                                                                                                                                                                                                                                                                                                                                                                                                                                                                                                                                                                                                                                                                                                                                                                                                                                                                                                                                                                                                                                                                                                                                                                                                                                                                                                                                                                                                                                                                                                                                                                                                                                                                                                                                                    |  |  |  |  |  |  |  |
|      |                                                                                                                                                                                                                                                                                                                                                                                                                                                                                                                                                                                                                                                                                                                                                                                                                                                                                                                                                                                                                                                                                                                                                                                                                                                                                                                                                                                                                                                                                                                                                                                                                                                                                                                                                                                                                                                                                                                                                                                                                                                                                                                                |  |  |  |  |  |  |  |
| 1035 | GONTIKAKI, E., MAYOR, D. J., THORNTON, B., BLACK, K. & WITTE, U. 2011. Processing of C-13-                                                                                                                                                                                                                                                                                                                                                                                                                                                                                                                                                                                                                                                                                                                                                                                                                                                                                                                                                                                                                                                                                                                                                                                                                                                                                                                                                                                                                                                                                                                                                                                                                                                                                                                                                                                                                                                                                                                                                                                                                                     |  |  |  |  |  |  |  |
| 1036 | labelled diatoms by a bathyal community at sub-zero temperatures Marine Ecology Progress Series 421 39-50                                                                                                                                                                                                                                                                                                                                                                                                                                                                                                                                                                                                                                                                                                                                                                                                                                                                                                                                                                                                                                                                                                                                                                                                                                                                                                                                                                                                                                                                                                                                                                                                                                                                                                                                                                                                                                                                                                                                                                                                                      |  |  |  |  |  |  |  |
| 2000 |                                                                                                                                                                                                                                                                                                                                                                                                                                                                                                                                                                                                                                                                                                                                                                                                                                                                                                                                                                                                                                                                                                                                                                                                                                                                                                                                                                                                                                                                                                                                                                                                                                                                                                                                                                                                                                                                                                                                                                                                                                                                                                                                |  |  |  |  |  |  |  |
| 1037 | GONTIKAKI, E., POLYMENAKOU, P. N., THORNTON, B., NARAYANASWAMY, B. E., BLACK, K.,                                                                                                                                                                                                                                                                                                                                                                                                                                                                                                                                                                                                                                                                                                                                                                                                                                                                                                                                                                                                                                                                                                                                                                                                                                                                                                                                                                                                                                                                                                                                                                                                                                                                                                                                                                                                                                                                                                                                                                                                                                              |  |  |  |  |  |  |  |
| 1038 | TSELEPIDES A & WITTE 11 2012 Microbial Response to Organic Matter Enrichment in the Oligotrophic                                                                                                                                                                                                                                                                                                                                                                                                                                                                                                                                                                                                                                                                                                                                                                                                                                                                                                                                                                                                                                                                                                                                                                                                                                                                                                                                                                                                                                                                                                                                                                                                                                                                                                                                                                                                                                                                                                                                                                                                                               |  |  |  |  |  |  |  |
| 1030 |                                                                                                                                                                                                                                                                                                                                                                                                                                                                                                                                                                                                                                                                                                                                                                                                                                                                                                                                                                                                                                                                                                                                                                                                                                                                                                                                                                                                                                                                                                                                                                                                                                                                                                                                                                                                                                                                                                                                                                                                                                                                                                                                |  |  |  |  |  |  |  |
| 1039 | Levantine Basin (Eastern Mediterranean). Geomicrobiology Journal, 29, 648-655.                                                                                                                                                                                                                                                                                                                                                                                                                                                                                                                                                                                                                                                                                                                                                                                                                                                                                                                                                                                                                                                                                                                                                                                                                                                                                                                                                                                                                                                                                                                                                                                                                                                                                                                                                                                                                                                                                                                                                                                                                                                 |  |  |  |  |  |  |  |
| 1040 | GONTIKAKI E. VAN OEVELEN D. SOETAERT K & WITTE II 2011 Food was flowe through a sub                                                                                                                                                                                                                                                                                                                                                                                                                                                                                                                                                                                                                                                                                                                                                                                                                                                                                                                                                                                                                                                                                                                                                                                                                                                                                                                                                                                                                                                                                                                                                                                                                                                                                                                                                                                                                                                                                                                                                                                                                                            |  |  |  |  |  |  |  |
| 1040 | state and the second states in the second states and st |  |  |  |  |  |  |  |
| 1041 | arctic deep-sea benthic community. Progress in Oceanography, 91, 245-259.                                                                                                                                                                                                                                                                                                                                                                                                                                                                                                                                                                                                                                                                                                                                                                                                                                                                                                                                                                                                                                                                                                                                                                                                                                                                                                                                                                                                                                                                                                                                                                                                                                                                                                                                                                                                                                                                                                                                                                                                                                                      |  |  |  |  |  |  |  |
| 1042 |                                                                                                                                                                                                                                                                                                                                                                                                                                                                                                                                                                                                                                                                                                                                                                                                                                                                                                                                                                                                                                                                                                                                                                                                                                                                                                                                                                                                                                                                                                                                                                                                                                                                                                                                                                                                                                                                                                                                                                                                                                                                                                                                |  |  |  |  |  |  |  |
| 1042 | HAKINEII, H. E., KEIL, K. G., HEDGES, J. I. & DEVOL, A. H. 1998. Influence of oxygen exposure time on                                                                                                                                                                                                                                                                                                                                                                                                                                                                                                                                                                                                                                                                                                                                                                                                                                                                                                                                                                                                                                                                                                                                                                                                                                                                                                                                                                                                                                                                                                                                                                                                                                                                                                                                                                                                                                                                                                                                                                                                                          |  |  |  |  |  |  |  |
| 1043 | organic carbon preservation in continental margin sediments. Nature, 391, 572-574.                                                                                                                                                                                                                                                                                                                                                                                                                                                                                                                                                                                                                                                                                                                                                                                                                                                                                                                                                                                                                                                                                                                                                                                                                                                                                                                                                                                                                                                                                                                                                                                                                                                                                                                                                                                                                                                                                                                                                                                                                                             |  |  |  |  |  |  |  |
|      |                                                                                                                                                                                                                                                                                                                                                                                                                                                                                                                                                                                                                                                                                                                                                                                                                                                                                                                                                                                                                                                                                                                                                                                                                                                                                                                                                                                                                                                                                                                                                                                                                                                                                                                                                                                                                                                                                                                                                                                                                                                                                                                                |  |  |  |  |  |  |  |
|      |                                                                                                                                                                                                                                                                                                                                                                                                                                                                                                                                                                                                                                                                                                                                                                                                                                                                                                                                                                                                                                                                                                                                                                                                                                                                                                                                                                                                                                                                                                                                                                                                                                                                                                                                                                                                                                                                                                                                                                                                                                                                                                                                |  |  |  |  |  |  |  |

**1012 BUHRING, S. I., LAMPADARIOU, N., MOODLEY, L., TSELEPIDES, A. & WITTE, U. 2006b. Benthic**

- HEDGES, J. I. & KEIL, R. G. 1995. Sedimentary organic matter preservation: an assessment and speculative
  synthesis. Marine Chemistry, 49, 81-115.
- HERMAN, P. M. J., MIDDELBURG, J. J., VAN DE KOPPEL, J. & HEIP, C. H. R. 1999.Ecology of estuarine
  macrobenthos.In: NEDWELL, D. B. & RAFFAELLI, D. G. (eds.) Advances in Ecological Research, Vol 29:
  Estuaries.
- HERMAN, P. M. J., MIDDELBURG, J. J., WIDDOWS, J., LUCAS, C. H. & HEIP, C. H. R. 2000. Stable
  isotopes' as trophic tracers: combining field sampling and manipulative labelling of food resources for
  macrobenthos. Marine Ecology-Progress Series, 204, 79-92.
- HUBAS, C., DAVOULT, D., CARIOU, T. & ARTIGAS, L. F. 2006. Factors controlling benthic metabolism
  during low tide along a granulometric gradient in an intertidal bay (Roscoff Aber Bay, France). Marine Ecology
  Progress Series, 316, 53-68.
- HUBAS, C., ARTIGAS, L. F. & DAVOULT, D. 2007. Role of the bacterial community in the annual benthic
  metabolism of two contrasted temperate intertidal sites (Roscoff Aber Bay, France). Marine Ecology Progress
  Series, 344, 39-48.
- HUETTEL, M., BERG, P. & KOSTKA, J. E. 2014.Benthic Exchange and Biogeochemical Cycling in
  Permeable Sediments.Annual Review of Marine Science, Vol 6, 6, 23-51.
- HUNTER, W. R., JAMIESON, A., HUVENNE, V. A. I. & WITTE, U. 2013.Sediment community responses to
   marine vs. terrigenous organic matter in a submarine canyon.Biogeosciences, 10, 67-80.
- HUNTER, W. R., LEVIN, L. A., KITAZATO, H. & WITTE, U. 2012-b. Macrobenthic assemblage structure
   and organismal stoichiometry control faunal processing of particulate organic carbo